# LLM-Explorer: A Plug-in Reinforcement Learning Policy Exploration Enhancement Driven by Large Language Models

**Qianyue Hao**, **Yiwen Song**, **Qingmin Liao, Jian Yuan, Yong Li**[†]
Department of Electronic Engineering, BNRist, Tsinghua University
Beijing, China

## Abstract

Policy exploration is critical in reinforcement learning (RL), where existing approaches include $\epsilon$-greedy, Gaussian process, etc. However, these approaches utilize preset stochastic processes and are indiscriminately applied in all kinds of RL tasks without considering task-specific features that influence policy exploration. Moreover, during RL training, the evolution of such stochastic processes is rigid, which typically only incorporates a decay in the variance, failing to adjust flexibly according to the agent's real-time learning status. Inspired by the analyzing and reasoning capability of large language models (LLMs), we design **LLM-Explorer** to adaptively generate task-specific exploration strategies with LLMs, enhancing the policy exploration in RL. In our design, we sample the learning trajectory of the agent during the RL training in a given task and prompt the LLM to analyze the agent's current policy learning status and then generate a probability distribution for future policy exploration. Updating the probability distribution periodically, we derive a stochastic process specialized for the particular task and dynamically adjusted to adapt to the learning process. Our design is a plug-in module compatible with various widely applied RL algorithms, including the DQN series, DDPG, TD3, and any possible variants developed based on them. Through extensive experiments on the Atari and MuJoCo benchmarks, we demonstrate LLM-Explorer's capability to enhance RL policy exploration, achieving an average performance improvement up to 37.27%. Our code is open-source at `https://github.com/tsinghua-fib-lab/LLM-Explorer` for reproducibility.

## 1 Introduction

In recent decades, reinforcement learning (RL) has been proven to be a powerful tool for training smart agents in solving sequential decision-making problems [1, 2]. The success of deep RL is especially noteworthy in tasks with high complexity, such as game playing [3, 4, 5, 6], chip design [7], smart city governance [8, 9, 10, 11, 12, 13, 14], where deep RL agents now exhibit performance surpassing human professionals in more and more scenarios. In the training of RL agents, policy exploration plays an indispensable role, which allows the agents to sample a diverse range of actions and uncover better strategies that may not be immediately apparent. The explore-exploit trade-off is a critical aspect of reinforcement learning, where agents must balance exploring new possibilities to improve long-term rewards and exploiting known strategies to maximize immediate gains.

---

[1]Qianyue Hao and Yiwen Song contribute equally to this work.
[2]Yong Li is the corresponding author, email: `liyong07@tsinghua.edu.cn`

39th Conference on Neural Information Processing Systems (NeurIPS 2025).

Various policy exploration approaches have been proposed in existing RL algorithms, including $\epsilon$-greedy in DQN [15], Gaussian process noise in DDPG [16], etc. Despite their success, existing methods lack adaptability and flexibility. First, they are designed based on preset stochastic processes applied uniformly across all kinds of tasks without any environment-specific adaption, neglecting the unique characteristics of different environments that may influence policy exploration. Besides, the evolution of these stochastic processes during training tends to be simplistic, which typically merely involves a gradual decay in variance over time. As a result, these methods fail to flexibly adjust the policy exploration strategy based on the agent's real-time learning status, potentially reducing the effectiveness of policy exploration, especially in complex or non-stationary environments.

There exist several challenges in addressing these limitations. First, RL tasks span diverse environments, and the training process involves many action steps, during which the agent's learning status undergoes complex changes. Thus, relying on more fine-grained manual designs based on preset stochastic processes becomes increasingly impractical. Moreover, given its widespread success, there have been many well-established RL algorithms with proven performance, and how to incorporate improvements into these existing methods to enhance policy exploration while preserving their original strengths requires investigation.

Facing these challenges, we propose to enhance policy exploration in RL based on LLMs, namely **LLM-Explorer**. The emerging LLMs, with advanced analyzing and reasoning capabilities [17, 18], demonstrate the potential to automatically analyze the environment characteristics and the agent's real-time learning status, thereby enabling more adaptive and flexible policy exploration. In LLM-Explorer, during the RL training process within a given environment, we periodically sample recent action-reward trajectories from the agent's experience and prompt the LLM to analyze the agent's current policy learning status based on the trajectories. The LLM then generates a tailored probability distribution that guides future policy exploration based on the agent's learning status and the specific characteristics of the environment. We update the probability distribution regularly, allowing it to dynamically adapt as the agent progresses through training and ensuring the exploration strategy evolves in response to changes in learning status. By doing so, we derive a specialized stochastic process from this dynamically updated distribution, which is uniquely suited to the environment. LLM-Explorer is designed to be a plug-in module that can be seamlessly integrated into existing RL algorithms by simply substituting the original preset stochastic process with the LLM-driven one without requiring any other architectural changes. Therefore, it is compatible with the DQN series [19, 20, 21, 22], DDPG [16], TD3 [23], as well as any possible variants developed based on them, making it a versatile solution for various RL tasks, covering both discrete and continuous action spaces. We conduct extensive experiments on the Atari [24, 25] and MuJoCo [26] benchmarks, and the results demonstrate the capability of LLM-Explorer.

In summary, the main contributions of this work include:

- We propose LLM-Explorer, a method that leverages LLMs to dynamically adjust the policy exploration during RL training in different tasks, which addresses the limitations of traditional policy exploration with preset stochastic processes.

- Our approach is designed as a plug-in module, allowing seamless integration with various widely applied RL algorithms, enabling enhanced exploration in both discrete and continuous action spaces without modifications to existing RL architectures.

- We conduct extensive experiments to evaluate our method in improving RL policy exploration across various tasks, attaining an average performance improvement up to 37.27%.

## 2 Problem Formulation

### 2.1 Markov Decision Process (MDP)

Markov decision process (MDP) is the fundamental framework for reinforcement learning, where an agent solves the decision-making problems in interaction with a dynamic environment. Mathematically, an MDP is defined by a tuple $(\mathcal{S}, \rho, \mathcal{A}, P, R)$ with $\mathcal{S}$ representing the state space, and $\rho \in \Delta(\mathcal{S})$ denoting the probability distribution of initial state, where $\Delta(\mathcal{S})$ is a collection of probability distribution over $\mathcal{S}$. $\mathcal{A}$ denotes the action space, and when executing a specific action in a given state, $P : \mathcal{S} \times \mathcal{A} \to \Delta(\mathcal{S})$ and $R : \mathcal{S} \times \mathcal{A} \to \mathbb{R}$ are the state transition probability function and the single-step reward function, respectively. At time step $t$, the agent executes action $a_t \in \mathcal{A}$

under the state of $s_t \in \mathcal{S}$, and then receives a reward of $r_t$ and experiences the state transition to $s_{t+1}$. The agent's goal in an MDP is to maximize its cumulative reward over time, which is the sum of discounted single-step rewards. This cumulative reward at time step $t$ is formalized as $G_t = \sum_{k=0}^{\infty} \gamma^k r_{t+k}$, where $\gamma$ is the discount factor that determines the importance of future rewards. To achieve this, the agent needs to balance exploiting known strategies and exploring unknown ones, where the former one means selecting the action with the largest estimated cumulative reward. In contrast, the latter requires trying other possibilities with randomness.

## 2.2 Large Language Models (LLMs)

Large language models are sophisticated neural networks with billions of parameters, which are mainly trained by predicting the probability of the next word in a sequence. Given $\{w_1, w_2, ..., w_{t-1}\}$, the models output $w_t$ to maximize the observation likelihood in the corpus as:

$$\prod_{t=1}^{T} P(w_t | w_1, w_2, ..., w_{t-1}). \tag{1}$$

Over the past few years, LLMs have made significant progress, where notable examples include the GPT family [27, 28, 29], the Llama family [30, 31], etc. These LLMs have exhibited strong capability across a wide range of natural language processing tasks, ranging from text generation and translation to summarization and question answering [17, 32, 33, 34, 35, 36].

# 3 Methods

## 3.1 Overview

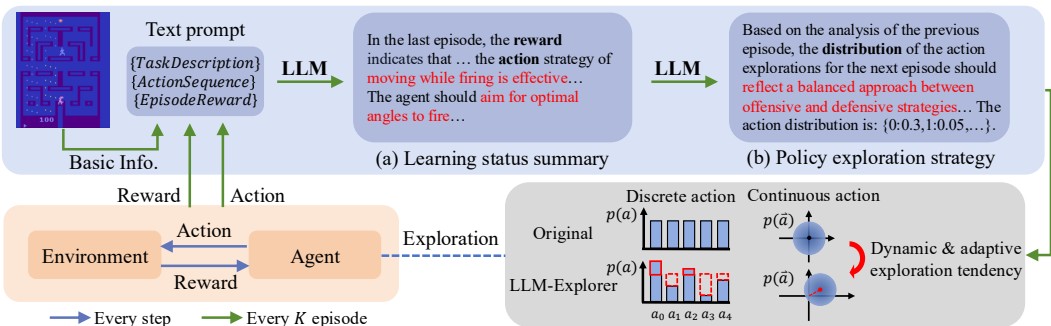

Figure 1: Illustration of LLM-Explorer, which utilizes LLMs to generate task-specialized stochastic processes that can be dynamically adjusted to adapt to the learning process, enhancing policy exploration in RL.

In this paper, we propose to improve the policy exploration in RL based on LLMs, namely **LLM-Explorer**. As shown in Figure 1, our framework employs two LLMs that collaborate through natural language communication and guide the policy exploration through a structured process. First, we introduce the basic task description and sample action-reward trajectories of the agent from the previous episode, prompting the former LLM to summarize the learning status of the agent and recommend potential exploration strategies (Section 3.2). Then, we feed the obtained summary and suggestion to the second LLM, which subsequently generates a probability distribution for policy exploration in the next $K$ episodes (Section 3.3). Here, $K$ is the hyper-parameter representing the interval at which the probability distribution is updated. Our method is a plug-in design that simply modifies the probability distribution for policy exploration in existing RL algorithms with the LLM-driven one without any other architectural changes, making it compatible with a wide range of RL algorithms covering both discrete and continuous action spaces (Section 3.4).

## 3.2 Learning Status Summarizing

To effectively guide the policy exploration, we design the first LLM to summarize the learning status of the agent every $K$ episode and provide suggestions on future exploration (Figure 1a). To achieve

this, we first describe the basic elements of the task as *{TaskDescription}*, ensuring that the LLM is aware of the tasks' characteristics.

**Task Description**: The task is a reinforcement learning problem where an agent *{TaskDetails}*. The action space is *{ActionDetails}*. The agent receives a reward of *{RewardDetails}*. The game ends when *{EndConditions}*. The goal is to *{GoalDetails}*.

Then, at each time of updating, we sample $M$ actions uniformly from the latest episode, obtaining *{ActionSequence}*, where $M$ stands for the sampling density. We also extract the total reward of the latest episode, obtaining *{EpisodeReward}*. Combining these, we design a tailored prompt for the first LLM, as formulated below:

**Prompt 1**: You are describing the last episode of the training process on a task. *{TaskDescription}*. In the last episode, the total reward is *{EpisodeReward}*, and the action sequence extracted at intervals is *{ActionSequence}*. Please analyze the data, generate a description, and provide possible strategy recommendations.

This prompt provides the necessary context for the LLM to summarize the information in the previous episode and extract insights into the agent's learning status. Additionally, it requires the LLM to offer potential strategy recommendations, aiming to provide more useful information for the upcoming policy exploration strategy generation.

It is worth mentioning that many RL tasks, such as the Atari benchmark, represent the environmental states by sequences of image frames, making it difficult for LLMs to process. In our design, we only sample the actions and rewards of the agent and exclude the states. The reason for this design is that our primary objective is to obtain an adaptive probability distribution for policy exploration based on the agent's learning status, rather than determining exact actions directly from the current state. Therefore, without requiring precise state information, LLMs can analyze what action patterns are most likely helpful for the current task from the task description and identify whether these action patterns have been adequately explored from the agent's historical behaviors and rewards. With such a design, our LLMs work with purely textual inputs, reducing computational consumption and ensuring compatibility with either multi-modal or text-only LLMs.

### 3.3 Policy Exploration Strategy Generation

To improve policy exploration, we design the second LLM in our framework to generate a probability distribution over the action space for future exploration (Figure 1b). This distribution is generated based on the first LLM's analysis regarding the learning status of the agent in the previous episode, as well as its suggestions for future policy exploration. We feed this information into the second LLM through the prompt structured as follows:

**Prompt 2**: You are determining the probability distribution for action exploration in reinforcement learning. *{TaskDescription}*. Here is a description of the situation in the previous episode: *{Summary&Suggestion}*. Based on the above information, please analyze what kind of actions should be selected to better improve the task effectiveness. *{OutputFormat}*.

With this prompt, the LLM analyzes what actions should be explored more frequently and outputs a probability distribution for the next $K$ episodes. This process enables the agent to prioritize actions that are more likely to improve the performance while also highlighting previously underexplored actions to discover new strategies. By periodically updating the strategy every $K$ episode, we ensure the policy exploration evolves dynamically to adapt to the agent's learning progress.

### 3.4 Compatibility with Different RL Algorithms

In order to make the LLM-Explorer compatible with RL algorithms for both discrete action and continuous action spaces, we design different methods for generating probability distributions for each type of action space. For discrete action spaces, we directly generate a probability distribution for selecting each possible action, replacing the uniform distribution in the original algorithms. The *{OutputFormat}* for policy exploration strategy generation is:

**Output Format (Discrete Action)**: Please output the distribution of the *{ActionNum}* action explorations for the next episode in decimal form. The format should be: {1: [probability], ...}.

For continuous action spaces, we generate a bias corresponding to the dimension of the action space and add it to the original symmetric Gaussian distribution centered around the origin, resulting in a biased probability distribution for action exploration with tendency. The *{OutputFormat}* is:

**Output Format (Continuous Action)**: The approach is to add a Gaussian noise to each dimension of action, and you need to decide the bias of the Gaussian noise for each dimension. Please output the bias for each of the *{ActionDim}* dimensions of actions for action explorations in the next episode based on your analysis in decimal form. Your output format should be: {1: [bias], 2: ...}.

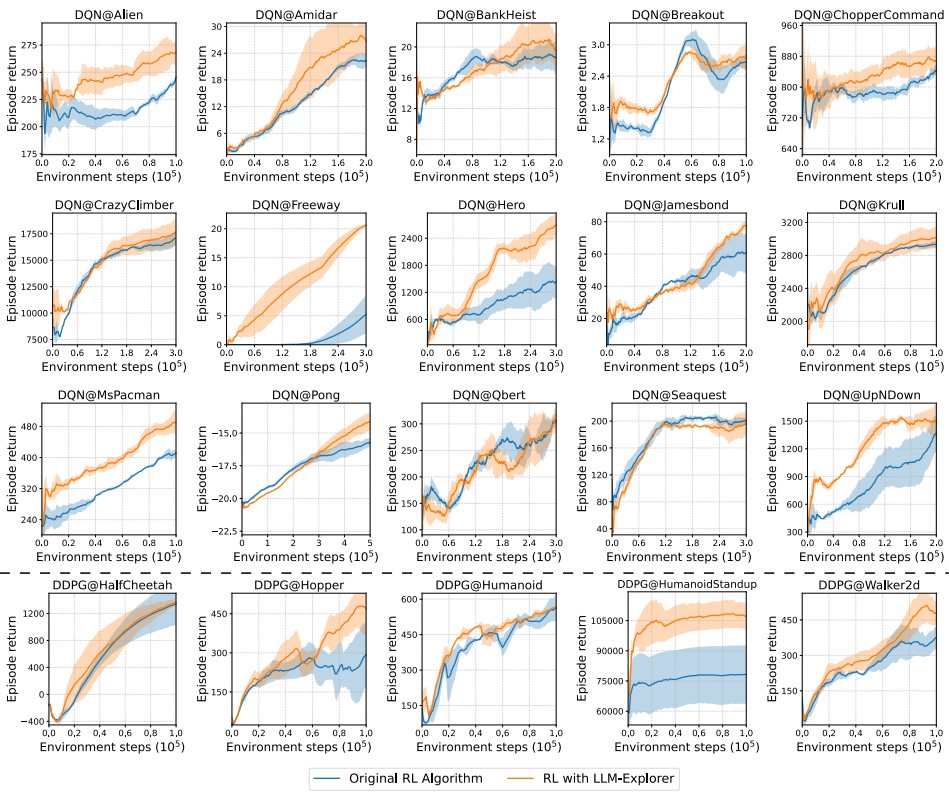

Figure 2: Performance of LLM-Explorer on the Atari and MuJoCo benchmark. In each experiment, we repeat with three different random seeds and the shaded areas indicate the standard deviations.

# 4 Experiments

## 4.1 Experimental Settings

We evaluate the performance of LLM-Explorer on the Atari [24, 25] and MuJoCo [26] benchmarks, covering tasks with both discrete and continuous action spaces. In the main experiments, we use DQN [15] and DDPG [16] as the basis on Atari and MuJoCo, respectively, and plug our LLM-Explorer into them. We selected 15 out of 26 tasks from Atari and 5 out of 11 from MuJoCo, where raw DQN or DDPG algorithm can converge stably and obtain good rewards. In addition, we set the number of training steps to 100k-500k across different tasks based on how fast the reward increases when training the original DQN or DDPG algorithm. In the LLM-Explorer module, we use GPT-4o mini[1] as the core LLM and set the two key parameters in our design, namely action sampling density and exploration adjusting interval, as $M = 100$ and $K = 1$. In our deployment, we fix a set of hyper-parameters across all environments. For reproducibility, we provide specific values of all hyper-parameters in Appendix A and list detailed contents of the prompts in Appendix L.

---

[1]`https://platform.openai.com/docs/models/gpt-4o-mini`

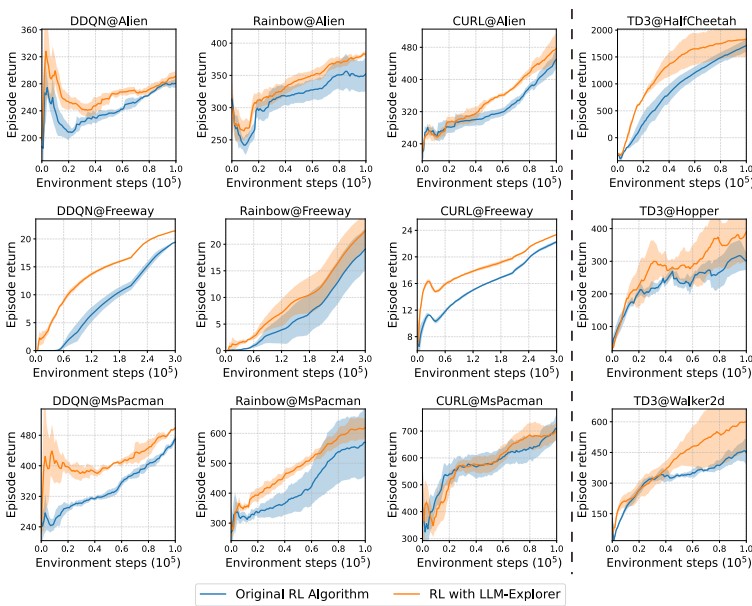

Figure 3: Compatibility of LLM-Explorer with various RL algorithms. In each experiment, we repeat with three different random seeds and use the shaded area to indicate the standard deviations.

## 4.2 Overall Performance

We train agents using the basis algorithm and algorithm with our LLM-Explorer module in the aforementioned environments, where in each environment, we repeat the training process with three different random seeds and average the results. We show the learning curves for each environment in Figure 2. On both the Atari and MuJoCo, LLM-Explorer improves the performance in most environments, verifying its ability to enhance the performance of the existing RL algorithm. Specifically, on the Atari tasks, LLM-Explorer reaches an increment of 37.27% and 13.84% on the mean and median human-normalized game score at the end of training [37, 38], as summarized in Appendix B.

Also, we compare our method with two commonly used exploration methods without LLMs in Appendix C. The results illustrate the advantage of our LLM-Explorer design, highlighting the benefit of introducing LLMs to enhance policy exploration.

In our design, LLM-Explorer is a simple plug-in method that can be seamlessly integrated with a wide range of existing RL algorithms. To verify, besides the basic algorithms aforementioned, we selected another three widely applied variants of DQN (see Appendix D), including Double-Dueling DQN [19, 20], Rainbow [21], and CURL [22]. We also include TD3 [23], a commonly used upgrade of DDPG. Respectively from the Atari and MuJoCo tasks, we selected three environments with relatively good training outcomes as representatives. In the selected tasks, we train agents with the original versions of the above RL algorithms, as well as the versions integrating our LLM-Explorer module. In each experiment, we repeat the training process with three different random seeds and average the results. We show the learning curves for the 12 experiments (4 algorithms×3 environments) in Figure 3. As the results illustrate, different RL algorithms exhibit diverse performance in different environments, while LLM-Explorer consistently improves their performance. This proves LLM-Explorer's compatibility with various RL algorithms, indicating its potential in tasks with either discrete or continuous action spaces.

## 4.3 Computational Overhead

In this section, we analyze the computational time and cost, which is crucial for assessing the practical application potential of our method. As shown in table 1, on average, each training run consumes approximately $1.3 in API calls and takes about 10 hours to complete. We consider this to be an acceptable and reasonable overhead. We also provided more detailed computational overhead comparison with baseline methods in Appendix F.

Table 1: Wall-clock time and API cost for a 500k-step run across all Atari and MuJoCo tasks. For each task, we average 3 repeated trainings with different random seeds.

| Environment | Task | Input token (k) | Output token (k) | Cost ($) | Time (h) |
|---|---|---|---|---|---|
| Atari | Alien | 1243.65 | 897.95 | 0.73 | 9.23 |
| | Amidar | 1563.54 | 927.00 | 0.79 | 9.37 |
| | BankHeist | 1514.98 | 1101.00 | 0.89 | 9.42 |
| | Breakout | 3821.06 | 2634.00 | 2.15 | 11.02 |
| | ChopperCommand | 1340.64 | 684.00 | 0.61 | 9.18 |
| | CrazyClimber | 413.23 | 208.00 | 0.19 | 8.59 |
| | Freeway | 366.87 | 231.25 | 0.19 | 8.58 |
| | Hero | 1191.81 | 686.00 | 0.59 | 9.12 |
| | Jamesbond | 2133.88 | 1011.00 | 0.93 | 9.64 |
| | Krull | 805.92 | 438.00 | 0.38 | 8.85 |
| | MsPacman | 1456.50 | 1006.10 | 0.82 | 9.36 |
| | Pong | 260.83 | 312.00 | 0.23 | 8.57 |
| | Qbert | 2263.39 | 1626.00 | 1.32 | 9.95 |
| | Seaquest | 1389.57 | 768.00 | 0.67 | 9.23 |
| | UpNDown | 1447.75 | 1011.00 | 0.82 | 9.36 |
| MuJoCo | HalfCheetah | 978.16 | 524.98 | 0.46 | 8.96 |
| | Hopper | 5693.25 | 3611.96 | 3.02 | 12.21 |
| | Humanoid | 25387.59 | 7477.76 | 8.29 | 22.03 |
| | HumanoidStandup | 2061.12 | 610.47 | 0.68 | 9.45 |
| | Walker2d | 4422.74 | 2475.20 | 2.15 | 11.21 |
| Average | | 2987.82 | 1412.08 | 1.30 | 10.17 |

## 4.4 Compatibility with Different LLMs

Table 2: Compatibility of LLM-Explorer with various LLMs. The human-norm scores (%) are recorded at the end of training and averaged across 3 random seeds. The underlines indicate improvements over the raw RL algorithm. The bold fonts are the best results.

| Task | DQN | DQN+LLM-Explorer | | | | |
|---|---|---|---|---|---|---|
| | | GPT-4o mini | GPT-4o | GPT-3.5 | Llama-3.1-405B | Llama-3.1-70B |
| Alien | 0.26 | 0.59 | 0.31 | 0.42 | **0.67** | 0.61 |
| Freeway | 17.75 | **69.71** | 67.27 | 66.45 | 60.22 | 63.7 |
| MsPacman | 1.56 | **2.75** | 1.63 | 1.53 | 1.88 | 2.01 |

In the framework work of LLM-Explorer, we utilize the LLMs with text-only prompts, leveraging their text-processing capability to derive smart policy exploration strategies. Instead of relying on some specific types or versions of LLMs, our design is a general framework that can work with various types of LLMs. To evaluate this, besides GPT-4o mini used above, we test several other LLMs that are most widely known, including GPT-4o[2], GPT-3.5[3], Llama-3.1-405B, and Llama-3.1-70B[4]. We train agents with the original DQN algorithm and then integrate DQN with our LLM-Explorer method, where the latter is driven by each of these different LLMs. In each experiment, we repeat the training process with three different random seeds and average the results. We summarize the game scores obtained at the end of training in Table 2 and show the learning curves in these experiments in Appendix E. In the results, our method consistently improves the human-normalized score of the original algorithms despite the type of LLMs, indicating its strong compatibility with different LLMs.

## 4.5 Balance between Performance and Consumption

To facilitate the wide application of our method, it is important to understand the relationship between its performance and computational consumption. Since LLM-Explorer is a simple plug-in design

---

[2]https://platform.openai.com/docs/models/gpt-4o
[3]https://platform.openai.com/docs/models
[4]https://ai.meta.com/blog/meta-llama-3-1

that does not impact the original computational consumption in RL training, we mainly focus on its auxiliary consumption in utilizing LLMs.

Table 3: Performance of LLM-Explorer with various ablation designs. The human-norm scores (%) are recorded at the end of training and averaged across 3 random seeds. The underlines indicate improvements over the raw RL algorithm. The bold fonts are the best results.

| Task | DQN | DQN+LLM-Explorer | | | | | | | | |
|---|---|---|---|---|---|---|---|---|---|---|
| | | Full design | | | w/o summarize & suggestion | | | w/o environment information | | |
| | | Score | Token in (k) | Token out (k) | Score | Token in (k) | Token out (k) | Score | Token in (k) | Token out (k) |
| Alien | 0.26 | **0.59** | 248.73 | 179.59 | 0.51 | 111.07 | 112.54 | 0.38 | 186.41 | 165.90 |
| Freeway | 17.75 | **69.71** | 220.12 | 138.75 | 68.97 | 88.91 | 69.94 | 61.26 | 164.38 | 134.93 |
| MsPacman | 1.56 | **2.75** | 291.30 | 201.22 | 2.32 | 129.18 | 125.22 | 1.89 | 222.05 | 208.31 |

**Components in the LLM workflow.** To uncover the roles of key components in the LLM workflow, we conduct ablation experiments. In one experiment, we remove the summarize & suggestion mechanism and allow a single LLM to directly output a probability distribution for future policy exploration based on the *{TaskDescription}*, *{ActionSequence}*, and *{EpisodeReward}*. In another experiment, we retain the two-stage design of the LLM workflow but do not provide the *{TaskDescription}*, only informing the LLMs of the environment's name. In each experiment, we repeat the training process with three different random seeds and average the results. As shown in Table 3 and Appendix E, both ablations continue to improve the performance of the original DQN algorithm while significantly reducing the token consumption of LLM. However, the first ablation lacks sufficient analysis of the agent's learning status, making it less flexible for adjustment during the training process. The second ablation lacks sufficient environmental information, making it less adaptive to specific environments. As a result, neither of them performs as well as the full design.

Table 4: Performance of LLM-Explorer with different action sampling density $M$ and exploration adjusting interval $K$. The human-norm scores (%) are recorded at the end of training and averaged across 3 random seeds. The underlines indicate improvements over the raw RL algorithm. The bold fonts are the best results.

| Task | DQN | DQN+LLM-Explorer | | | |
|---|---|---|---|---|---|
| | | $M$=100,$K$=1 | $M$=50,$K$=1 | $M$=100,$K$=2 | $M$=200,$K$=1 |
| Alien | 0.26 | 0.59 | 0.51 | 0.38 | **0.83** |
| Freeway | 17.75 | **69.71** | 64.72 | 66.52 | 66.52 |
| MsPacman | 1.56 | **2.75** | 2.22 | 2.07 | 2.24 |

**Setting of key parameters.** There are two key parameters within the workflow, namely action sampling density ($M$) and exploration adjusting interval ($K$). By reducing sampling density, i.e., smaller $M$, or reducing the frequency of adjusting the exploration strategy, i.e., larger $K$, we can obviously reduce the token consumption of LLM. To evaluate the impact of these, we conduct experiments and show the results in Table 4 and Appendix E. As the results illustrate, LLM-Explorer with either smaller $M$ or larger $K$ keeps improving the performance of the original DQN algorithm. However, smaller $M$ provides insufficient information about the agent's real-time learning status, and larger $K$ limits adjustments to the exploration strategy. As a result, both of them are less capable of flexibly adapting the policy exploration to the training process, achieving worse performance than LLM-Explorer with the original settings of $M$ and $K$. Moreover, we also analyze the impact of increasing the sampling density, i.e., larger $M$. As the results indicate, although increasing the token consumption of LLM, a larger $M$ does not consistently improve the performance of LLM-Explorer. This may be because the original settings of $M$ already provide sufficient information about the agent's real-time learning status. Therefore, further increasing the sampling density complicates the LLM's ability to analyze and summarize the data, which may hinder overall performance.

**Size of LLMs.** As shown above, the proposed LLM-Explorer framework is compatible with different LLMs. To further explore the impact of LLM size, we test multiple Qwen2.5 models with varying sizes. As shown in Table 5, the performance degrades with smaller models, and models with ≤7B

Table 5: Performance of LLM-Explorer with different LLM sizes. The human-norm scores (%) are recorded at the end of training and averaged across 3 random seeds. The underlines indicate improvements over the raw RL algorithm. The bold fonts are the best results.

| Task | DQN | DQN+LLM-Explorer | | | |
|---|---|---|---|---|---|
| | | Qwen2.5-32B | Qwen2.5-14B | Qwen2.5-7B | Qwen2.5-3B |
| Alien | 0.26 | **0.51** | 0.34 | 0.16 | 0.06 |
| Freeway | 17.75 | **59.49** | 30.78 | 26.32 | 16.93 |
| MsPacman | 1.56 | **2.60** | 1.94 | 1.51 | 1.14 |

parameters generally fail to work. This suggests that, in general, larger and more capable models tend to perform better.

From these analyses, we demonstrate that the full design pipeline, properly configured values of $M$ and $K$, and capable large LLMs are critical for achieving the best performance of LLM-Explorer. However, we also highlight the trade-offs between performance and computational consumption in LLM-Explorer. Therefore, for deployments with limited computational resources, it is possible to simplify the design of the LLM workflow or adjust $M$ and $K$ as above to reduce computational consumption while still maintaining certain performance improvements over the original RL algorithm. For deployments with sufficient computational resources, the full design with the original settings of $M$ and $K$ is the optimal choice. Also, fine-tuning a specialized LLM, for example, distilling the exploration strategy generation from a large LLM into a smaller one, would further reduce the minimum capable LLM size and save more computational cost.

## 4.6 Case Studies

To demonstrate the rationality in determining the policy exploration strategy with LLM-Explorer, we provide an intuitive case study in Figure 4 within the environment of the Freeway. In this environment, the goal is crossing the busy road safely, while the action space includes three items, namely no-ops, moving up, and moving down. In case 1, the previous action of the agent involves a large proportion of 'no ops', and the LLM in the stage of learning status summarizing points out its overly caution behavior that lacks clear direction. Subsequently, the latter LLM generates an exploration strategy that stresses moving up and down. In case 2, the previous action of the agent involves a large proportion of 'moving up', and the former LLM reveals that the current learning status of the agent is actively aiming to reach the other side of the highway. Based on this, the latter LLM generates an exploration strategy that further encourages 'moving up' to reach the goal while also adding a small proportion of 'moving down' to adjust position relative to traffic for safety. Such rational analyses enable our design to generate smart policy exploration strategies that are adaptive to specific environments and learning processes, enhancing the performance of various RL algorithms.

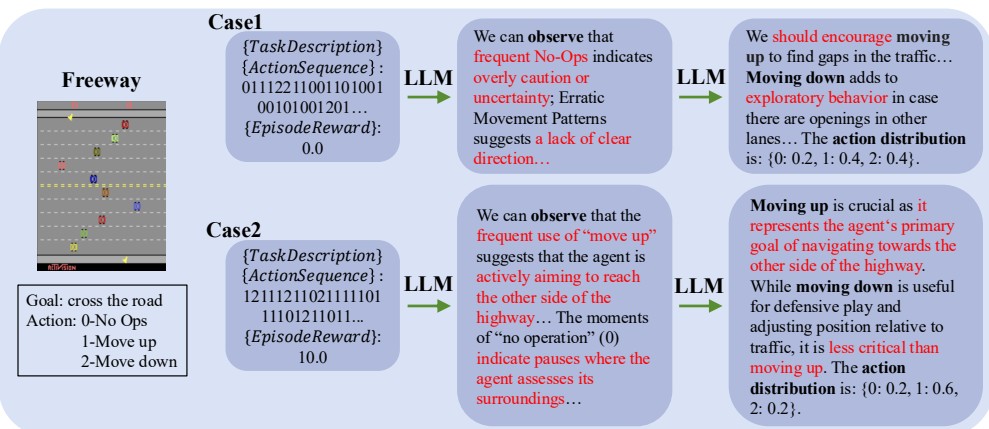

Figure 4: Case study of the operating process of LLM-Explorer.

# 5 Related Works

## 5.1 Policy Exploration in RL

Plentiful approaches have been used in existing RL algorithms for policy exploration. One of the most basic methods is the $\epsilon$-greedy strategy used in DQN [15], where with a probability of $\epsilon$, the agent randomly samples an action from all possible actions rather than greedily exploiting the current best one. As an improvement of DQN, Noisy-DQN introduces noisy networks [39], which inject randomness directly into the action selection process, allowing for better policy exploration. Other methods utilize the randomness introduced by Gaussian distributions. For example, the actions are sampled from Gaussian distributions in PPO [40], and small Gaussian noises are added to the deterministic actions in DDPG [16]. Also, in some implementations of DDPG [41, 42, 43], the standard white Gaussian noise is replaced with an Ornstein-Uhlenbeck (OU) process with temporal correlation [44, 45], leading to smoother and potentially more effective policy exploration. Moreover, extensive algorithms incorporate an entropy term in the reward function [46, 47, 48], encouraging more diverse action selections to enhance policy exploration. However, these methods are designed based on preset stochastic processes, which can neither adapt to specific environments nor be flexibly adjusted during the training process. In contrast, we design to dynamically generate a stochastic process by LLMs to guide policy exploration, which is adaptive and flexible.

## 5.2 Enhancing RL with LLMs

Many studies have explored the use of LLMs in enhancing the performance of RL [49]. First, a significant body of work focuses on leveraging LLMs to design reward functions based on the characteristics of the tasks, providing feedback for the agent's policy learning [50, 51, 52, 53]. Additionally, other research investigates using LLMs to design state representation functions, offering more effective state inputs for the agents [54]. On a macro level, LLMs have been utilized to decompose complex tasks into sub-goals [50] or provide high-level instructions [55] to facilitate RL training. Moreover, LLMs are employed in human-AI coordination, enabling humans to specify the desired strategies for RL agents through natural language instructions [56]. Despite these works, it remains unknown how to leverage LLMs to enhance policy exploration in RL, where this paper manages to bridge such a knowledge gap. Furthermore, our plug-in module can integrate with a wide range of existing works using LLMs to enhance RL from various aspects, further benefiting their performance from the aspect of policy exploration.

# 6 Conclusions

In this paper, we propose a compatible plug-in design that utilizes LLMs to enhance policy exploration in RL algorithms. We design to use LLMs to analyze the agent's real-time learning status based on its action-reward trajectory and then periodically update the probability distribution for policy exploration. By doing so, we are able to adapt the policy exploration to any specific task and flexibly adjust it during the training process, only with the requirement of low-cost text-only prompts. Through extensive experiments and in-depth analyses in various environments, we verify the validity of our design and illustrate its compatibility with a wide range of established RL algorithms, covering tasks with both discrete and continuous action spaces.

## Acknowledgments and Disclosure of Funding

This work was supported in part by the National Key Research and Development Program of China under 2024YFC3307500, and Beijing National Research Center for Information Science and Technology.

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

# A   Implementation Details

In this section, we provide the main implementation details for reproducibility in Table 6. Please refer to our source code at `https://github.com/tsinghua-fib-lab/LLM-Explorer` for the exact usage of each hyper-parameters and more details.

Table 6: Implementation details.

| Module | Element | Detail |
|---|---|---|
| System | OS | Ubuntu 22.04.2 |
| | CUDA | 11.7 |
| | Python | 3.11.4 |
| | Device | 8*NVIDIA A100 80G |
| DQN and variants | $\gamma$ | 0.99 |
| | Batch Size | 256 |
| | Interval of target network updating | 1000 |
| | Optimizer | Adam |
| | Learning rate | 0.0001 |
| | Replay buffer size | 10000 |
| | Start epsilon | 1 |
| | Min epsilon | 0.1 |
| | Epsilon decay per step | 0.99999 |
| DDPG and TD3 | $\gamma$ | 0.99 |
| | Batch Size | 256 |
| | Optimizer | Adam |
| | Learning rate of actor | 0.00001 |
| | Learning rate of critic | 0.0001 |
| | Replay buffer size | 10000 |
| | $\tau$ of target network updating | 1000 |
| | $\sigma$ for the exploration noise | 0.1 |
| | (only TD3) $\sigma$ for the policy noise | 0.2 |
| | (only TD3) Policy delay | 2 |
| | (only TD3) Update iteration | 10 |
| Learning status summarizing | Model name | gpt-4o-mini-2024-07-18 |
| | Temperature | 1.0 |
| Policy exploration strategy generation | Model name | gpt-4o-mini-2024-07-18 |
| | Temperature | 1.0 |
| Test of different LLMs | Model name for GPT-4o | gpt-4o-2024-08-06 |
| | Temperature for GPT-4o | 1.0 |
| | Model name for GPT-3.5 | gpt-3.5-turbo-0125 |
| | Temperature for GPT-3.5 | 1.0 |
| | Model name for Llama-3.1-405B | Llama-3.1-405B-Instruct |
| | Temperature for Llama-3.1-405B | 1.0 |
| | Model name for Llama-3.1-70B | Llama-3.1-70B-Instruct |
| | Temperature for Llama-3.1-70B | 1.0 |

# B    Scores on Atari games

Here, we summarize the Atari game scores obtained at the end of training in Table 7. To better compare the games with varying score ranges and difficulty levels, we also normalize the game scores using the average score of human players [37, 38]. The human-norm score is calculated as:

$$human - norm\ score = \frac{score_{agent} - score_{random}}{score_{human} - score_{random}} \qquad (2)$$

Table 7: Performance of LLM-Explorer on the Atari benchmark, where the results are recorded at the end of training and averaged across 3 random seeds. The bold fonts indicate the best results.

| Task | DQN | | DQN+LLM-Explorer | | Improvement (%) |
|---|---|---|---|---|---|
| | Score | Human-norm score (%) | Score | Human-norm score (%) | |
| Alien | 245.46 | 0.26 | 268.44 | **0.59** | 126.92 |
| Amidar | 22.34 | 0.97 | 26.75 | **1.22** | 25.77 |
| BankHeist | 18.64 | 0.6 | 19.51 | **0.72** | 20.00 |
| Breakout | 2.67 | 3.36 | 2.74 | **3.62** | 7.74 |
| ChopperCommand | 840.63 | 0.45 | 868.33 | **0.87** | 93.33 |
| CrazyClimber | 17070.76 | 25.11 | 17694.35 | **27.6** | 9.92 |
| Freeway | 5.25 | 17.75 | 20.64 | **69.71** | 292.73 |
| Hero | 1439.7 | 1.38 | 2689.62 | **5.58** | 304.35 |
| Jamesbond | 60.84 | 11.63 | 77.35 | **17.66** | 51.85 |
| Krull | 2933.05 | 125.06 | 3009.12 | **132.19** | 5.70 |
| MsPacman | 411.07 | 1.56 | 489.9 | **2.75** | 76.28 |
| Pong | -15.71 | 14.13 | -14.13 | **18.61** | 31.71 |
| Qbert | 306.07 | **1.07** | 301.97 | 1.04 | -2.80 |
| Seaquest | 201.58 | **3.18** | 196.15 | 3.05 | -4.09 |
| UpNDown | 1370.99 | 7.51 | 1489.54 | **8.57** | 14.11 |
| Total-Mean | 1660.89 | 14.27 | 1809.35 | **19.59** | 37.27 |
| Total-Median | 245.46 | 3.18 | 268.44 | **3.62** | 13.84 |

The results indicate that LLM-Explorer improves the human-normalized score in 13 out of 15 environments, with an increment of 37.27% and 13.84%, respectively, on the mean and median score, verifying its ability to enhance the performance of the existing RL algorithm.

# C   Baseline Comparisons

To illustrate the advantage of our LLM-Explorer design, we compare our method with two widely used common exploration methods without LLMs:

- **NoisyNet** [39]: Deep reinforcement learning agent with parametric noise added to its weights. The induced stochasticity of the agent's policy can be used to aid efficient exploration.
- **Random network distillation (RND)** [57]: An exploration bonus for deep reinforcement learning methods that is the error of a neural network predicting features of the observations given by a fixed randomly initialized neural network. This bouns encourages the exploration of unfamiliar states.

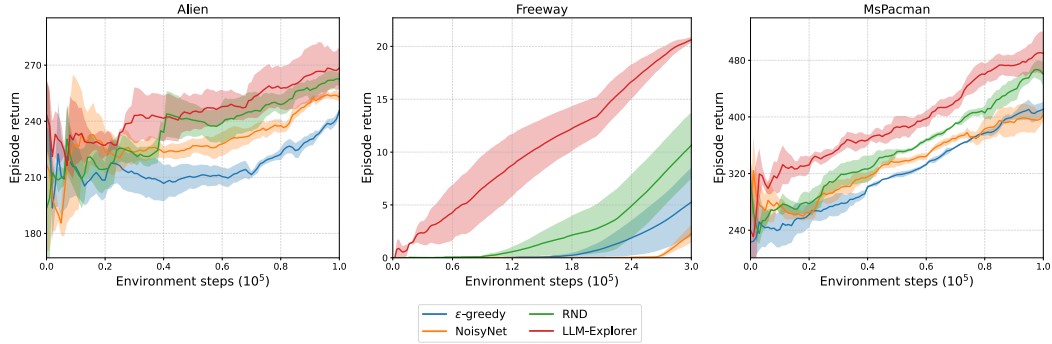

Figure 5: Comparison among our LLM-Explorer design with baselines. In each experiment, we repeatedly run the training process with three different random seeds and use the shaded area to indicate the standard deviations.

We train RL models in the Atari environments of Alien, Freeway, and MsPacman, and show the training process in Figure 5. We first train with the original DQN algorithms, and then integrate it with NoisyNet, RND, and our LLM-Explorer method, respectively. In the results, our method consistently outperforms the baseline methods without LLMs, indicating the advantage of our design, which utilizes LLMs to generate stochastic process specialized for the policy exploration in a particular task and dynamically adjust the tendency of exploration to adapt to the learning process.

# D   Deep Q-Learning and Its Variants

One of the most established methods for solving RL tasks is the Deep Q Networks algorithm [15], which trains a neural network $Q_\theta$ to approximate the agent's action-reward mapping. DQN updates the parameters of $Q_\theta$ by minimizing the error between predicted reward from $Q_\theta$ and its greedily estimated target value:

$$\mathcal{L}_\theta^{DQN} = \left( Q_\theta(s_t, a_t) - \left( r_t + \gamma \max_{a'} Q_\theta\left(s_{t+1}, a'\right) \right) \right)^2 . \tag{3}$$

Specifically in DQN, policy exploration is achieved by the $\epsilon$-greedy mechanism, where most of the time, the agent executes $a_t$ that maximizes $(Q_\theta(s_t, a_t)$, while with a small probability of $\epsilon$, the agent randomly selects $a_t$ from the action space.

Various improvements have been made to improve the original DQN. Prioritized experience replay [58] improves data efficiency by adding importance sampling into the replaying buffer. Double-DQN [19] modifies the target value, namely $(r_t + \gamma \max_{a'} Q_\theta(s_{t+1}, a'))$, by substituting $Q_\theta$ with the target network $Q_{\theta'}$, which is a delayed copy of $Q_\theta$ to avoid overestimation. Dueling-DQN [20] improves the network structure of $Q_\theta$ to decouple the state value from the advantage of taking a given action in that state. Noisy-DQN [39] introduces noisy networks, which inject randomness directly into the network of $Q_\theta$, allowing for better policy exploration. Ultimately, Rainbow [21] consolidates these improvements into a single combined algorithm, and CURL [22] enhances the performance of Rainbow by adding an unsupervised contrastive learning target.

# E    Learning curves

Here, we show the learning curves in the experiments of the main texts.

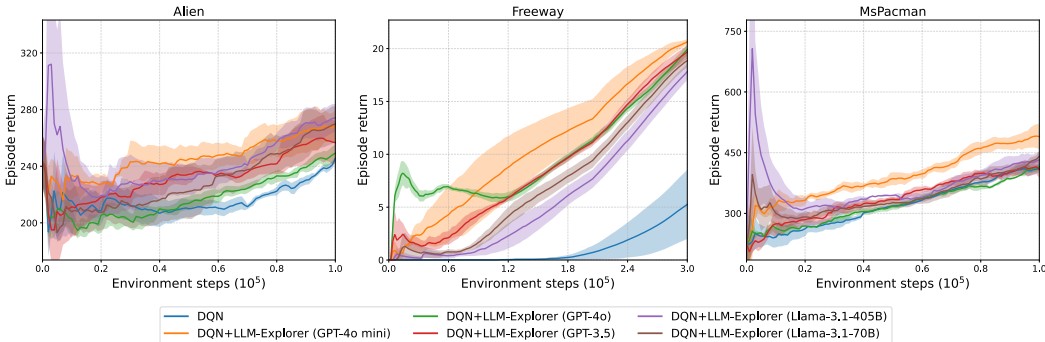

Figure 6: Compatibility of LLM-Explorer with various LLMs. In each experiment, we repeatedly run the training process with three different random seeds and use the shaded area to indicate the standard deviations.

In Figure 6, we show the training process with the original DQN algorithm and then integrate it with our LLM-Explorer method, where the latter is driven by different LLMs. In the results, our method consistently improves the performance despite the type of LLMs, indicating its strong compatibility with different LLMs.

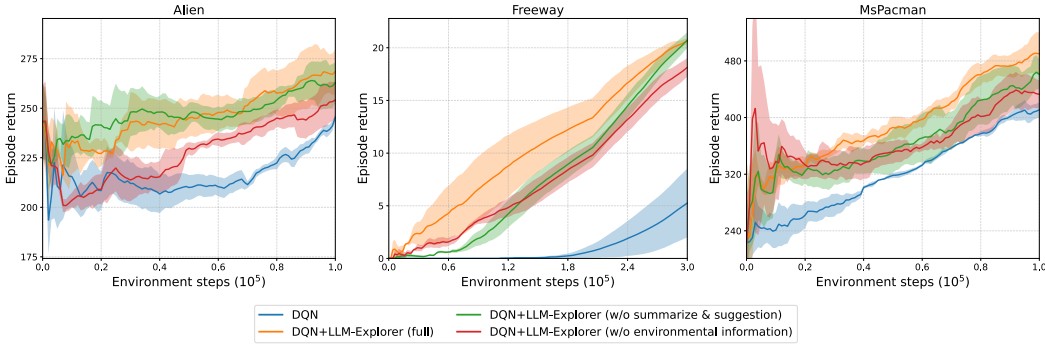

Figure 7: Performance of LLM-Explorer with various ablation designs. In each experiment, we repeatedly run the training process with three different random seeds and use the shaded area to indicate the standard deviations.

In Figure 7, we show the training process with the original DQN algorithm and then integrate it with our LLM-Explorer method, where the latter contains different ablation designs. In the results, both ablations continue to improve the performance of the original DQN algorithm while significantly reducing the token consumption of LLM. However, the first ablation lacks sufficient analysis of the agent's learning status, making it less flexible for adjustment during the training process. The second ablation lacks sufficient environmental information, making it less adaptive to specific environments. As a result, neither of them performs as well as the full design of LLM-Explorer.

In Figure 8, we show the training process with the original DQN algorithm and then integrate it with our LLM-Explorer method, where the latter is configured with different values of $M$. In the results, LLM-Explorer with smaller $M$ keeps improving the performance of the original DQN algorithm. However, smaller $M$ provides insufficient information about the agent's real-time learning status, achieving worse performance than LLM-Explorer with the original settings of $M$.

In Figure 9, we show the training process with the original DQN algorithm and then integrate it with our LLM-Explorer method, where the latter is configured with different values of $K$. In the results, LLM-Explorer with larger $K$ keeps improving the performance of the original DQN algorithm.

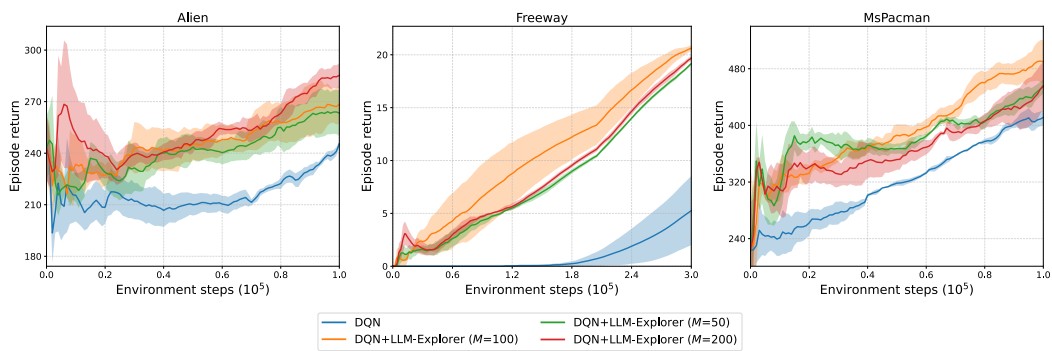

Figure 8: Performance of LLM-Explorer with different action sampling density $M$. In each experiment, we repeatedly run the training process with three different random seeds and use the shaded area to indicate the standard deviations.

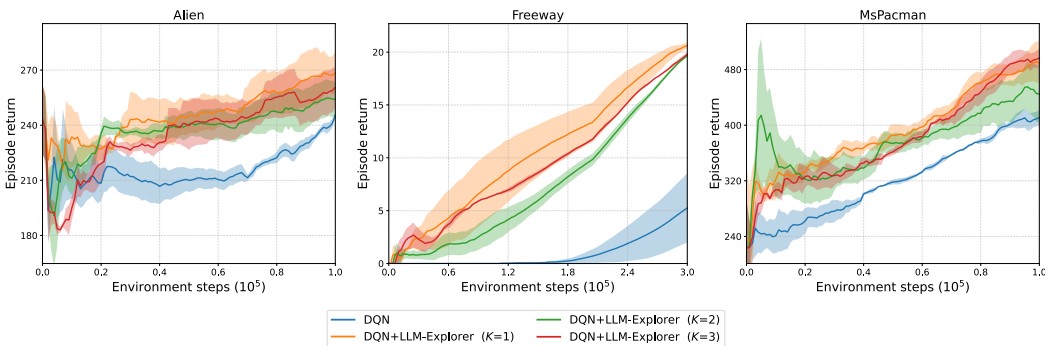

Figure 9: Performance of LLM-Explorer with different exploration adjusting interval $K$. In each experiment, we repeatedly run the training process with three different random seeds and use the shaded area to indicate the standard deviations.

However, larger $K$ limits adjustments on the exploration strategy, achieving worse performance than LLM-Explorer with the original settings of $K$.

In all the figures above, we repeat the training process with three different random seeds in each experiment and average the results. We use the shaded area to indicate the standard deviations.

# F   Computational Overhead Comparison

To provide a more direct comparison with baselines under a fixed time budget, we conduct an additional experiment against NoisyNet. We count the time overhead used for the LLM API call in each update cycle of LLM-Explorer to allow the NoisyNet agent to perform additional training rollouts and updates. This ensures both methods operate within the same total wall-clock time.

Table 8: Comparison of LLM-Explorer and NoisyNet under a fixed time budget. Both methods operate within the same total wall-clock time. The scores are recorded at the end of training and averaged across 3 random seeds.

| Task | Time (h) | LLM-Explorer | | NoisyNet | |
|---|---|---|---|---|---|
| | | Steps (k) | Score | Steps (k) | Score |
| Alien | 9.23 | 500 | 414.79 | 554 | 377.64 |
| Freeway | 8.58 | 500 | 29.16 | 515 | 10.39 |
| MsPacman | 9.36 | 500 | 1173.19 | 562 | 879.12 |

As shown in Table 8, even when the NoisyNet baseline is allowed to train for more iterations, LLM-Explorer maintains a clear performance advantage. This shows that our design, calling LLM once every $K$ episodes, can well balance efficiency and performance.

# G Impact of Input Trajectory Length

Our current method updates the exploration strategy based on $M$ actions sampled uniformly from the single latest episode. Intuitively, providing data from more historical episodes could give the LLM a richer context for its analysis. To see what will actually happen, we test 2, 3, and 5 episodes.

Table 9: Performance and token usage for different input trajectory length of LLM-Explorer. The scores are recorded at the end of training and averaged across 3 random seeds.

| Task | DQN | DQN+LLM-Explorer | | | | | | | |
|---|---|---|---|---|---|---|---|---|---|
| | | 1 episode | | 2 episode | | 3 episode | | 5 episode | |
| | | Token in (k) | Score | Token in (k) | Score | Token in (k) | Score | Token in (k) | Score |
| Alien | 245.26 | 248.73 | 268.44 | 277.73 | 273.19 | 306.73 | 275.18 | 364.73 | 267.49 |
| Freeway | 5.25 | 220.12 | 20.64 | 249.32 | 21.26 | 278.52 | 21.44 | 336.92 | 20.73 |
| MsPacman | 411.07 | 291.30 | 489.90 | 326.30 | 501.71 | 361.30 | 505.68 | 431.30 | 481.30 |

As We find that performance does improve when using more historical data, but there is a bottleneck. The best performance is achieved when using data from approximately 3 episodes. When providing too many episodes, it may result in an input sequence that is overly long and complex, which can hinder the LLM's ability to effectively process the information. More inputs also lead to a proportional increase in computational cost.

# H  Adaptive Update Interval

Our existing results have shown that simply reducing the update frequency leads to a decline in performance in Table 3. Here, we design a simple heuristic mechanism to trigger LLM updates more adaptively. After each episode, we calculate the average score improvement over the last 5 episodes ($g_1$) and the last 10 episodes ($g_2$). If $g_1$ is lower than $g_2$ by more than a given threshold $G$, we infer that the current exploration strategy may be losing effectiveness and trigger an LLM update.

Table 10: Performance of LLM-Explorer with different exploration strategy update frequency. The scores are recorded at the end of training and averaged across 3 random seeds.

| Task | DQN | DQN+LLM-Explorer | | | | | | | |
|------|-----|------------------|------|------|------|------|------|------|------|
| | | Fix K=1 | | Fix K=2 | | Adaptive G=10% | | Adaptive G=30% | |
| | | Avg. interval | Score | Avg. interval | Score | Avg. interval | Score | Avg. interval | Score |
| Alien | 245.26 | 1.00 | 268.44 | 2.00 | 254.02 | 1.51 | 265.19 | 1.86 | 261.43 |
| Freeway | 5.25 | 1.00 | 20.64 | 2.00 | 19.16 | 1.64 | 20.51 | 1.98 | 19.82 |
| MsPacman | 411.07 | 1.00 | 489.90 | 2.00 | 454.80 | 1.47 | 486.29 | 1.74 | 479.18 |

As shown in Table 10, the adaptive mechanism reduced average update frequency. Also, the resulting performance degradation is less severe than a fixed update interval change. For future research, more sophisticated adaptive triggers are valuable to improve the performance-cost balance.

# I Automatic Task Description

To improve the automation of our framework, we design to employ an additional LLM to automatically generate the task descriptions following the steps of:

1. We first state the required format of the task description in Section 3.2 and manually craft a task description for the 'Alien' game like Appendix L.
2. We then prompt GPT-4o mini to generate task descriptions for other environments with the one-shot example.
3. We replace the original, hand-written task descriptions with the automatically generated ones and re-run the experiments.

**Prompts for automatic task description generation:** *You are writing a task description for {TaskName} to support the subsequent task solving. The required format is {TaskDescriptionFormat}. Here is an example for the Alien game from the Atari benchmark: {TaskDescriptionExample}. Please strictly follow the format and cover all details of actions, rewards, end conditions, and goals.*

To further investigate the role and necessity of this specific template, we conduct three additional experiments:

- **Only one-shot:** We prompt the LLM to generate task descriptions by providing only the manually crafted description for the 'Alien' game as a one-shot example, without including the specific template format itself.
- **Only template:** We prompt the LLM to generate task descriptions by providing only the manually crafted description for the 'Alien' game as a one-shot example, without including the specific template format itself.
- **Zero-shot:** We prompt the LLM to generate descriptions without providing either the specific template or the one-shot example.

**Prompts for automatic task description generation (only one-shot):** *You are writing a task description for {TaskName} to support the subsequent task solving. Here is an example for the Alien game from the Atari benchmark: {TaskDescriptionExample}. Please cover all details of actions, rewards, end conditions, and goals.*

**Prompts for automatic task description generation (only template):** *You are writing a task description for {TaskName} to support the subsequent task solving. The required format is {TaskDescriptionFormat}. Please strictly follow the format and cover all details of actions, rewards, end conditions, and goals.*

**Prompts for automatic task description generation (zero-shot):** *You are writing a task description for {TaskName} to support the subsequent task solving. Please cover all details of actions, rewards, end conditions, and goals.*

As shown in Table 11, LLM-Explorer with the auto-generated task descriptions performs similarly to the original one. This means that generating task descriptions can be automated without a significant drop in performance, which strengthens the scalability of our method to a broader range of tasks.

Also, using an one-shot example with a template, and using an one-shot example alone, yield similar results; both approaches allow LLM-Explorer to achieve performance comparable to using the original manual prompts. Using only the template is slightly less effective, while providing neither a template nor an example leads to a drop in performance. This analysis underscores that the structured details of actions, rewards, end conditions, and goals, as organized in our manual prompts, are crucial for LLM-Explorer's effectiveness. By checking the automated task descriptions generated under each setting, we find that providing either an one-shot example or a template is sufficient to guide the LLM to include the necessary task information. Providing a concrete, well-structured example proves to be slightly more effective than providing only an abstract template. Conversely, when the LLM is not given any reference and asked to generate a description from scratch, it cannot reliably include all the necessary information, which in turn harms the effectiveness of LLM-Explorer.

Table 11: Performance of LLM-Explorer with automatic task description generation. The scores are recorded at the end of training and averaged across 3 random seeds.

| Task | DQN | DQN+LLM-Explorer | | | | |
|---|---|---|---|---|---|---|
| | | Manual | Auto | Auto (only one-shot) | Auto (only template) | Auto (zero-shot) |
| Alien | 245.46 | 268.44 | – | – | – | – |
| Amidar | 22.34 | 26.75 | 25.16 | 25.41 | 23.93 | 23.12 |
| BankHeist | 18.64 | 19.51 | 19.60 | 19.21 | 19.24 | 18.87 |
| Breakout | 2.67 | 2.74 | 2.71 | 2.75 | 2.72 | 2.69 |
| ChopperCommand | 840.63 | 868.33 | 873.98 | 871.25 | 851.45 | 838.71 |
| CrazyClimber | 17070.76 | 17694.35 | 17731.42 | 17709.53 | 17521.82 | 17215.43 |
| Freeway | 5.25 | 20.64 | 19.41 | 19.88 | 18.95 | 8.31 |
| Hero | 1439.70 | 2689.62 | 2513.76 | 2498.62 | 2351.17 | 1688.93 |
| Jamesbond | 60.48 | 77.35 | 78.13 | 77.58 | 73.81 | 63.29 |
| Krull | 2933.05 | 3009.12 | 3013.67 | 3021.55 | 2988.67 | 2949.34 |
| MsPacman | 411.07 | 489.90 | 501.13 | 495.74 | 473.55 | 428.18 |
| Pong | -15.71 | -14.13 | -14.07 | -14.21 | -14.48 | -15.35 |
| Qbert | 306.07 | 301.97 | 303.53 | 301.19 | 302.58 | 305.15 |
| Seaquest | 201.58 | 196.15 | 202.03 | 195.87 | 197.32 | 199.86 |
| UpNDown | 1370.99 | 1489.54 | 1476.72 | 1467.13 | 1463.15 | 1394.51 |
| Average | 1761.99 | 1919.42 | 1910.51 | 1906.54 | 1876.71 | 1794.36 |

## J    Failure Modes and Safeguards

As the experimental results indicate, LLM-Explorer is not universally effective and its performance is limited in certain tasks like Qbert and Seaquest.

### J.1    Skewed Outputs of LLMs

First, we consider that one possible reason is that the LLM outputs extremely skewed distributions in some cases, which harm the training. To diagnose, we calculate the KL divergence of each distribution generated during training to a uniform distribution and examine the deviation between the maximum and average values.

Table 12: KL divergence statistics for various tasks. The results are recorded at the end of training and averaged across 3 random seeds.

| Task | Action dimension | $Max[KL(p,U)]$ | $\frac{Max[KL(p,U)] - Mean[KL(p,U)]}{std[KL(p,U)]}$ |
|---|---|---|---|
| Alien | 18 | 0.773 | 8.298 |
| Amidar | 10 | 1.205 | 5.037 |
| BankHeist | 18 | 2.082 | 12.477 |
| Breakout | 4 | 0.436 | 5.985 |
| ChopperCommand | 18 | 1.969 | 13.319 |
| CrazyClimber | 9 | 1.034 | 10.104 |
| Freeway | 3 | 0.58 | 4.458 |
| Hero | 18 | 0.847 | 7.385 |
| Jamesbond | 18 | 1.085 | 8.552 |
| Krull | 18 | 1.969 | 11.585 |
| MsPacman | 9 | 1.182 | 4.751 |
| Pong | 6 | 1.026 | 1.85 |
| **Qbert** | **6** | **1.670** | **15.015** |
| **Seaquest** | **18** | **2.085** | **16.906** |
| UpNDown | 6 | 0.399 | 6.810 |

As shown in Table 12, it reveals that the two tasks with the worst performance indeed exhibit a much greater deviation, confirming the more skewed distributions. These extreme distributions might push the exploration policy too aggressively and hinder the learning. Consider the characteristics of these tasks: they are both highly dynamic, featuring numerous moving objects and complex interactions. Such complexity makes it challenging for LLMs to obtain clear analyses.

Facing this, we design a simple safeguard mechanism to mitigate this. Before applying the generated exploration distributions, we calculate its KL divergence to uniform. If this value exceeds the running average of past KL divergences by more than 10 standard deviations, the agent discards the skewed distribution for that cycle and falls back to:

- **KL clip (uniform):** the default uniform exploration strategy.
- **KL clip (old):** the distribution from the last cycle.

Table 13: Performance of LLM-Explorer with KL clip safeguards. The scores are recorded at the end of training and averaged across 3 random seeds.

| Task | DQN | DQN+LLM-Explorer | | | | | |
|---|---|---|---|---|---|---|---|
| | | Original | | KL clip (uniform) | | KL clip (old) | |
| | | Score | $Max[KL(p,U)]$ | Score | $Max[KL(p,U)]$ | Score | $Max[KL(p,U)]$ |
| Qbert | 306.07 | 301.97 | 1.670 | 312.43 | 1.038 | 316.92 | 1.104 |
| Seaquest | 201.58 | 196.15 | 2.085 | 209.37 | 1.234 | 221.81 | 1.427 |

As shown in Table 13, the safeguard design can partially mitigate the instability, and the mechanism that reuses the last valid strategy, achieves better performance. Our current method for identifying over-skewed distributions, which is based on KL divergence, is a preliminary and simple approach.

Designing more sophisticated mechanisms to more reliably identify exploration distributions that are detrimental to training remains a valuable and important area for future research.

## J.2 Hallucination of LLMs

Second, the intrinsic hallucination of LLMs can cause the model to generate irrational exploration distribution in some cases. To improve, we design a simple mitigation strategy inspired by self-consistency work in LLM reasoning [59]. During each update cycle, we use the same prompt to have the LLM generate an exploration distribution twice. Due to the randomness introduced by the temperature setting, this process yields two different distributions. We then calculate the KL divergence to uniform for both outputs and select the distribution with the smaller KL divergence.

Table 14: Performance of LLM-Explorer with self-consistency safeguards. The scores are recorded at the end of training and averaged across 3 random seeds.

| Task | DQN | DQN+LLM-Explorer | | | |
| --- | --- | --- | --- | --- | --- |
| | | Original | | Self-consistency) | |
| | | Score | $Max[KL(p, U)]$ | Score | $Max[KL(p, U)]$ |
| Qbert | 306.07 | 301.97 | 1.670 | 309.21 | 1.122 |
| Seaquest | 201.58 | 196.15 | 2.085 | 205.12 | 1.360 |

As shown in Table 14, this approach demonstrates a slightly positive effect on performance. To better counter the negative impacts of intrinsic LLM hallucination within the LLM-Explorer framework, adapting other sophisticated methods from the field of LLM reasoning, such as self-reflection [60], presents another potential direction for future research.

## J.3 Randomness in the Initial State

Third, the randomness in the initial states of the environments may impact the performance of LLM-Explorer. Let's imagine an extreme task: "get to the center of the room." If in the first episode, the agent is randomly initialized on the left side, then "moving right" actions are beneficial. But if the next episode starts on the right, the LLM, being unaware of this new initial state, would still favor exploring "move right" based on the last episode's experience.

However, most tasks, including those in our benchmarks, are not as extreme as this hypothetical example. Our state-agnostic approach works based on two main factors:

- **Structure similarity of initial states.** While the initial positions of elements like the agent, obstacles (e.g., cars in Freeway), and enemies (e.g., ghosts in MsPacman) are randomized, their overall structure and relationships between these elements are generally consistent across episodes.

- **Dynamic similarity of agent-environment interactions.** Most tasks involve long and complex processes, not just simple, few-step action sequences. The underlying dynamics of how the agent interacts with the environment are largely the same regardless of the specific starting positions.

Thereby, the effective action distribution tends to have strong similarities across different episodes of the same task. LLM-Explorer can analyze these general patterns based on the task description and action-reward history to improve the exploration distribution. For instance, as shown in our case study of Freeway in Section 4.6, the agent may start at different positions on the bottom of road and car patterns may vary, but to achieve the goal of "reaching the top of the road", the action "moving up" is almost always beneficial, while "no-ops" is generally ineffective. The LLM can correctly infer this state-agnostic strategic bias.

That said, if a task truly matches the extremity of the hypothetical example we imagined above, our current LLM-Explorer framework would likely perform poorly. For such cases, the framework can be extended to incorporate a VLM, which would receive the initial state images as an additional input alongside the task descriptions. For example, if the Freeway task were modified so the goal is to

"reach the opposite side" and the agent could start at either the top or the bottom, the VLM would need the initial state to determine whether "moving up" or "moving down" should be prioritized.

In general, for most normal tasks, the current state-agnostic design is a good choice considering computational cost, while for some extreme situations, a VLM version may be required to incorporate initial state information.

# K  State Awareness and LLM's Prior Knowledge about Tasks

In this section, we discuss the role of the task description in the prompt. Given that a task description is provided in the prompt in Section 3.2, one might question whether the LLM truly understands the characteristics of the task and generates an appropriate probability distribution for policy exploration or if it has merely memorized which actions work well in each environment. We present two pieces of evidence that provide insight into this question.

First, under our standard design, the task description includes detailed information about the actions and goals but does not include the task's name. Experimental results presented in Section 4.5 show that when all information is removed and the prompt only includes the task name, performance declines. If the LLM had simply memorized which actions work well in each environment, providing only the task name would allow the model to more easily recognize the task and retrieve the corresponding actions from its memory, leading to better performance. However, the observed performance drop suggests that the LLM requires a detailed analysis and understanding of the task description rather than relying on the task name to retrieve memorized action strategies.

Second, in some tasks, such as those with well-established strategies, the required actions are relatively clear. In contrast, other tasks, like the MsPacman game, require more localized and fine-grained actions without an obvious pre-existing strategy. Despite this, our method performs well on such tasks, indicating that the LLM is able to analyze and understand the task's detailed description rather than simply retrieving memorized strategies for known tasks.

Together, these findings suggest that, in our design, the LLM genuinely understands the task's characteristics and generates a suitable probability distribution for policy exploration. This enables the model to perform effectively on new tasks, even when no prior knowledge of what actions work well is available.

# L Detailed prompts

Here we list the detailed *{TaskDescription}* in the prompts for Atari environments and MuJoCo environments.

## L.1 Atari Environments

- **Alien**: *The task is a reinforcement learning problem where an agent controls an astronaut navigating through a dangerous alien world. The action space is discrete with 18 options: {0: no operation, 1: fire, 2: move up, 3: move right, 4: move left, 5: move down, 6: move up-right, 7: move up-left, 8: move down-right, 9: move down-left, 10: move up and fire, 11: move right and fire, 12: move left and fire, 13: move down and fire, 14: move up-right and fire, 15: move up-left and fire, 16: move down-right and fire, 17: move down-left and fire}. In the environment, the agent receives +50 points for defeating an alien and +100 points for clearing a level. Small rewards like +10 points are given for collecting power-ups, while penalties include -50 points for taking damage and -100 points for losing a life. The game ends when the agent loses all lives, with the goal being to maximize cumulative rewards through effective combat, exploration, and survival.*

- **Amidar**: *The task is a reinforcement learning problem where an agent controls a character navigating a maze to avoid enemies and complete objectives by marking sections of the maze. The action space is discrete with 10 options: {0: no operation, 1: fire, 2: move up, 3: move right, 4: move left, 5: move down, 6: move up and fire, 7: move right and fire, 8: move left and fire, 9: move down and fire}. In the environment, the fire action has no functional effect, as the primary objective is to move through the maze. The observation space consists of raw pixel values representing the game screen, showing the character, enemies, and the maze layout. The agent receives +10 points for marking a section of the maze and +50 points for completing an entire maze level. Additionally, the agent earns +100 points for capturing an enemy while in a powered-up state, and +20 points for collecting special bonus items scattered throughout the environment. However, the agent is penalized with -50 points for being caught by an enemy, and an additional -5 points for excessive inaction or idling for too long. The game ends when the agent loses all lives or completes the entire maze. The goal is to maximize the score by navigating the maze efficiently while avoiding enemies.*

- **BankHeist**: *The task is a reinforcement learning problem where an agent controls a character involved in a bank heist, navigating through a dynamic environment filled with guards and obstacles. The action space is discrete with 18 options: {0: no operation, 1: fire, 2: move up, 3: move right, 4: move left, 5: move down, 6: move up-right, 7: move up-left, 8: move down-right, 9: move down-left, 10: move up and fire, 11: move right and fire, 12: move left and fire, 13: move down and fire, 14: move up-right and fire, 15: move up-left and fire, 16: move down-right and fire, 17: move down-left and fire}. The observation space consists of raw pixel values representing the game screen, showing the agent, guards, and loot. In this environment, the agent receives rewards for successfully stealing loot and evading or neutralizing guards. The game ends when the agent loses all lives, and the primary objective is to maximize cumulative rewards through stealthy navigation, effective shooting, and strategic interactions with the environment.*

- **Breakout**: *The task is a reinforcement learning problem where an agent controls a paddle at the bottom of the screen, aiming to hit a ball and break bricks at the top. The action space is discrete with 4 options: {0: no operation, 1: fire (launch the ball), 2: move right, 3: move left}. The observation space consists of raw pixel values representing the game screen, displaying the paddle, the ball, and the bricks. The reward mechanism is designed to incentivize the destruction of bricks, with the agent earning points each time a brick is broken. In this reward mechanism, players score points by hitting bricks of various colors with a ball. Each brick color is assigned a specific point value: red and orange bricks yield 7 points, yellow and green bricks grant 4 points, while aqua and blue bricks provide 1 point each. The game ends when the agent loses all its lives by failing to catch the ball with the paddle. The primary objective is to maximize cumulative rewards by strategically controlling the paddle to keep the ball in play and target higher-value bricks while avoiding misses.*

- **ChopperCommand**: *The task is a reinforcement learning problem where an agent controls a helicopter navigating through a desert environment filled with enemy vehicles and aircraft. The action space is discrete with 18 options: {0: no operation, 1: fire, 2: move up, 3: move right, 4: move left, 5: move down, 6: move up-right, 7: move up-left, 8: move down-right, 9: move down-left, 10: move up and fire, 11: move right and fire, 12: move left and fire, 13: move down and fire, 14: move up-right and fire, 15: move up-left and fire, 16: move down-right and fire, 17: move down-left and fire}. The observation space consists of raw pixel values representing the game screen, displaying the helicopter, enemy vehicles, aircraft, and fuel depots. In this reward design mechanism, players earn points by shooting down enemy aircraft: 100 points for each enemy helicopter and 200 points for each enemy jet. A bonus is awarded for destroying an entire wave of hostile aircraft, calculated by multiplying the number of remaining trucks in the convoy by the wave number (from one to ten) and then by 100. This system incentivizes players to maximize their score through both individual kills and strategic gameplay. The game ends when the agent runs out of fuel or is hit by enemy fire and loses all lives. The primary objective is to maximize cumulative rewards by skillfully navigating the environment, destroying enemies, collecting fuel, and avoiding hazards to survive as long as possible.*

- **CrazyClimber**: *The task is a reinforcement learning problem where an agent controls a climber scaling the side of a tall building while avoiding various obstacles. The action space is discrete with 9 options: {0: no operation, 1: move up, 2: move right, 3: move left, 4: move down, 5: move up-right, 6: move up-left, 7: move down-right, 8: move down-left}. The observation space consists of raw pixel values representing the game screen, displaying the climber, the building, windows, and various obstacles such as falling objects. In the reward mechanism, players earn points in two ways: climbing points for each row of windows climbed and bonus points for reaching the top of each skyscraper. The climbing points vary by building, with 100 points per row for Building 1, 200 for Building 2, 300 for Building 3, and 400 for Building 4. Bonus points serve as a timer; they start at a maximum value when climbing a new building and decrease by 100 points every ten seconds. To retain bonus points, players must reach the top and grab the helicopter within 30 seconds, as bonus points continue to decline until the helicopter is reached. The maximum bonus points also increase with each building, ranging from 100,000 points for Building 1 to 400,000 points for Building 4. The game ends when the climber falls or loses all lives. The primary objective is to maximize cumulative rewards by skillfully navigating the vertical environment, dodging hazards, and climbing as high as possible without falling.*

- **Freeway**: *The task is a reinforcement learning problem where an agent controls a character attempting to cross a busy highway filled with fast-moving cars. The action space is discrete with 3 options: {0: no operation, 1: move up, 2: move down}. The observation space consists of raw pixel values representing the game screen, displaying the character, various lanes of traffic, and the road. The reward mechanism is designed to incentivize the successful crossing of the highway. The agent earns points for reaching the other side of the road, with each successful crossing awarding a fixed number of points. There are no explicit negative rewards, but the agent loses time and progress when hit by a car, as it is sent back to the starting point. The game ends when a time limit is reached. The primary objective is to maximize cumulative rewards by skillfully navigating through the traffic, avoiding cars, and making as many successful crossings as possible before time runs out.*

- **Hero**: *The task is a reinforcement learning problem where an agent controls a hero navigating through an underground cave system filled with enemies and obstacles. The action space is discrete with 18 options: {0: no operation, 1: fire, 2: move up, 3: move right, 4: move left, 5: move down, 6: move up-right, 7: move up-left, 8: move down-right, 9: move down-left, 10: move up and fire, 11: move right and fire, 12: move left and fire, 13: move down and fire, 14: move up-right and fire, 15: move up-left and fire, 16: move down-right and fire, 17: move down-left and fire}. The observation space consists of raw pixel values representing the game screen, showing the hero, enemies, environmental hazards, and collectible items. The reward mechanism is designed to incentivize the exploration of the cave and the collection of various items, such as treasure. The agent earns points for defeating enemies and gathering treasures scattered throughout the cave. The hero may also gain points by rescuing trapped miners. There are penalties for losing health due to enemy attacks or environmental hazards. The game ends when all lives are lost. The primary objective is to maximize cumulative*

*rewards by skillfully navigating the cave system, defeating enemies, avoiding hazards, and collecting valuable items.*

- **Jamesbond**: *The task is a reinforcement learning problem where an agent controls James Bond navigating through various action-packed levels filled with enemies and obstacles. The action space is discrete with 18 options: {0: no operation, 1: fire, 2: move up, 3: move right, 4: move left, 5: move down, 6: move up-right, 7: move up-left, 8: move down-right, 9: move down-left, 10: move up and fire, 11: move right and fire, 12: move left and fire, 13: move down and fire, 14: move up-right and fire, 15: move up-left and fire, 16: move down-right and fire, 17: move down-left and fire}. The observation space consists of raw pixel values representing the game screen, displaying James Bond, various enemies, vehicles, and obstacles. In this reward system, players earn points by collecting various targets. For the reward system, each target has the following point value: a Diamond is worth 50 points, while the Frogman, Space Shuttle, and Submarine each provide 200 points. The Poison Bomb and Torpedo are worth 100 points each. The Spinning Satellite offers the highest reward at 500 points, while the Rapid Rocket and Fire Bomb also contribute 100 points each. Completing the mission yields a substantial bonus of 5,000 points. This design encourages players to explore actively and prioritize collecting high-value targets to maximize their cumulative score. The game ends when all lives are lost. The primary objective is to maximize cumulative rewards by skillfully navigating the levels, shooting enemies, and strategically completing missions while avoiding hazards and enemy attacks.*

- **Krull**: *The task is a reinforcement learning problem where an agent controls a character navigating through a vibrant fantasy world filled with enemies, moving platforms, and obstacles. The action space is discrete with 18 options: {0: no operation, 1: fire, 2: move up, 3: move right, 4: move left, 5: move down, 6: move up-right, 7: move up-left, 8: move down-right, 9: move down-left, 10: move up and fire, 11: move right and fire, 12: move left and fire, 13: move down and fire, 14: move up-right and fire, 15: move up-left and fire, 16: move down-right and fire, 17: move down-left and fire}. The observation space consists of raw pixel values representing the game screen, displaying the character, various enemies, laser barriers, and collectible items such as gems and keys. The reward mechanism is designed to incentivize progressing through different rooms by collecting keys to unlock doors and defeating enemies with laser shots. The agent earns points for defeating enemies, collecting gems, and clearing levels. The game becomes progressively more difficult with more enemies and complex rooms to navigate. The game ends when all lives are lost or when the player completes all levels. The primary objective is to maximize cumulative rewards by skillfully navigating the environment, defeating enemies, avoiding hazards, and collecting items to progress through the world.*

- **MsPacman**: *The task is a reinforcement learning problem where an agent controls Ms. Pacman navigating through a maze filled with pellets, power-ups, and enemy ghosts. The action space is discrete with 9 options: {0: no operation, 1: move up, 2: move right, 3: move left, 4: move down, 5: move up-right, 6: move up-left, 7: move down-right, 8: move down-left}. The observation space consists of raw pixel values representing the game screen, displaying Ms. Pacman, pellets, power pellets, and ghosts moving around the maze. The reward mechanism is designed to incentivize the collection of pellets and the strategic use of power-ups. Ms. Pacman earns points for each pellet collected and additional points for eating ghosts after consuming a power pellet. However, if she gets caught by a ghost without the power-up, a life is lost. The game ends when all lives are lost or when all pellets in the maze are collected. The primary objective is to maximize cumulative rewards by skillfully navigating the maze, avoiding or chasing ghosts when appropriate, and collecting as many pellets and power-ups as possible.*

- **Pong**: *The task is a reinforcement learning problem where an agent controls a paddle to hit a ball and score points by getting the ball past the opponent's paddle. The action space is discrete with 6 options: {0: no operation, 1: fire, 2: move the paddle up, 3: move the paddle down, 4: right fire, 5: left fire}. In the environment, the fire action has no functional effect, as we can only move the paddle up and down. The observation space consists of raw pixel values representing the game screen. The agent receives a reward of +1 for scoring and -1 when the opponent scores. The game ends when either side reaches 21 points.*

- **Qbert**: *The task is a reinforcement learning problem where an agent controls Qbert, a character navigating through a pyramid of cubes while avoiding enemies and hazards. The*

*action space is discrete with 6 options: {0: no operation, 1: fire (jump), 2: move up, 3: move right, 4: move left, 5: move down}. The observation space consists of raw pixel values representing the game screen, displaying Qbert, enemies, and the pyramid of cubes that Qbert must jump on to change their color. The reward mechanism is designed to incentivize jumping on cubes and avoiding enemies. Qbert earns points for each successful jump that changes the color of a cube, and additional points for completing a level by changing all cubes to the desired color. Penalties occur if Qbert is hit by enemies or falls off the pyramid, resulting in a lost life. The game ends when all lives are lost. The primary objective is to maximize cumulative rewards by skillfully navigating the pyramid, changing the colors of cubes, avoiding enemies, and completing levels efficiently.*

- **Seaquest**: *The task is a reinforcement learning problem where an agent controls a submarine navigating through an underwater world filled with enemy submarines, divers, and obstacles. The action space is discrete with 18 options: {0: no operation, 1: fire, 2: move up, 3: move right, 4: move left, 5: move down, 6: move up-right, 7: move up-left, 8: move down-right, 9: move down-left, 10: move up and fire, 11: move right and fire, 12: move left and fire, 13: move down and fire, 14: move up-right and fire, 15: move up-left and fire, 16: move down-right and fire, 17: move down-left and fire}. The observation space consists of raw pixel values representing the game screen, displaying the submarine, enemies, friendly divers, and the underwater environment. The reward mechanism is designed to incentivize the destruction of enemy submarines and the rescue of divers. The agent earns points for shooting enemy submarines and other hostile underwater threats, as well as for rescuing divers and bringing them safely to the surface. Penalties occur if the submarine is hit by enemy fire or runs out of oxygen, which results in a loss of life. The game ends when all lives are lost. The primary objective is to maximize cumulative rewards by skillfully navigating the underwater environment, avoiding enemies, rescuing divers, and managing oxygen levels effectively.*

- **UpNDown**: *The task is a reinforcement learning problem where an agent controls a car navigating through a colorful, fast-paced world filled with other vehicles and obstacles on winding roads. The action space is discrete with 6 options: {0: no operation, 1: fire, 2: move up, 3: move down, 4: move up and fire, 5: move down and fire}. The observation space consists of raw pixel values representing the game screen, displaying the agent's car, other vehicles, and road obstacles. The reward mechanism is designed to incentivize avoiding collisions and overtaking other vehicles. The agent earns points for passing other cars on the road and avoiding crashes. Higher rewards are earned by overtaking more cars and successfully navigating tricky sections of the road. The game ends when the agent collides with another car or falls off the road, resulting in a loss of life. The primary objective is to maximize cumulative rewards by skillfully maneuvering the car, avoiding collisions, overtaking as many vehicles as possible, an1'd progressing through the levels without losing lives.*

## L.2  MuJoCo Environments

- **HalfCheetah**: *The task is a reinforcement learning problem where an agent controls a 3-dimensional quadruped robot consisting of a torso (free rotational body) with four legs attached to it, where each leg has two body parts. The action space consists of 8 continuous values, each between -1 and 1, representing the torque applied at one hinge joints: {0: the rotor between the torso and back right hip, 1: the rotor between the back right two links, 2: the rotor between the torso and front left hip, 3: the rotor between the front left two links, 4: the rotor between the torso and front right hip, 5: the rotor between the front right two links, 6: the rotor between the torso and back left hip, 7: the rotor between the back left two links}. The goal is to coordinate the four legs to move in the forward (right) direction by applying torque to the eight hinges connecting the two body parts of each leg and the torso (nine body parts and eight hinges).*

- **Hopper**: *The task is a reinforcement learning problem where an agent controls a 2-dimensional one-legged figure consisting of four main body parts - the torso at the top, the thigh in the middle, the leg at the bottom, and a single foot on which the entire body rests. The action space consists of 3 continuous values, each between -1 and 1, representing the torque applied at one hinge joints: {0: the thigh rotor, 1: the leg rotor, 2: the foot rotor}.*

*The goal is to make the robot that move in the forward (right) direction by applying torque to the three hinges that connect the four body parts.*

- **Humanoid**: *The task is a reinforcement learning problem where an agent controls a 3-dimensional bipedal robot that is designed to simulate a human. It has a torso (abdomen) with a pair of legs and arms, and a pair of tendons connecting the hips to the knees. The legs each consist of three body parts (thigh, shin, foot), and the arms consist of two body parts (upper arm, forearm). The action space consists of 17 continuous values, each between -0.4 and 0.4, representing the torque applied at one hinge joints: {0: the hinge in the y-coordinate of the abdomen, 1: the hinge in the z-coordinate of the abdomen, 2: the hinge in the x-coordinate of the abdomen, 3: the rotor between torso/abdomen and the right hip (x-coordinate), 4: the rotor between torso/abdomen and the right hip (z-coordinate), 5: the rotor between torso/abdomen and the right hip (y-coordinate), 6: the rotor between the right hip/thigh and the right shin, 7: the rotor between torso/abdomen and the left hip (x-coordinate), 8: the rotor between torso/abdomen and the left hip (z-coordinate), 9: the rotor between torso/abdomen and the left hip (y-coordinate), 10: the rotor between the left hip/thigh and the left shin, 11: the rotor between the torso and right upper arm (coordinate-1), 12: the rotor between the torso and right upper arm (coordinate-2), 13: the rotor between the right upper arm and right lower arm, 14: the rotor between the torso and left upper arm (coordinate-1), 15: the rotor between the torso and left upper arm (coordinate-2), 16: the rotor between the left upper arm and left lower arm}. The goal of the task is to walk forward as fast as possible without falling over.*

- **HumanoidStandup**: *The task is a reinforcement learning problem where an agent controls a 3-dimensional bipedal robot that is designed to simulate a human. It has a torso (abdomen) with a pair of legs and arms, and a pair of tendons connecting the hips to the knees. The legs each consist of three body parts (thigh, shin, foot), and the arms consist of two body parts (upper arm, forearm). The action space consists of 17 continuous values, each between -0.4 and 0.4, representing the torque applied at one hinge joints: {0: the hinge in the y-coordinate of the abdomen, 1: the hinge in the z-coordinate of the abdomen, 2: the hinge in the x-coordinate of the abdomen, 3: the rotor between torso/abdomen and the right hip (x-coordinate), 4: the rotor between torso/abdomen and the right hip (z-coordinate), 5: the rotor between torso/abdomen and the right hip (y-coordinate), 6: the rotor between the right hip/thigh and the right shin, 7: the rotor between torso/abdomen and the left hip (x-coordinate), 8: the rotor between torso/abdomen and the left hip (z-coordinate), 9: the rotor between torso/abdomen and the left hip (y-coordinate), 10: the rotor between the left hip/thigh and the left shin, 11: the rotor between the torso and right upper arm (coordinate -1), 12: the rotor between the torso and right upper arm (coordinate -2), 13: the rotor between the right upper arm and right lower arm, 14: the rotor between the torso and left upper arm (coordinate -1), 15: the rotor between the torso and left upper arm (coordinate -2), 16: the rotor between the left upper arm and left lower arm}. The goal of the task is to make the humanoid stand up and then keep it standing by applying torques to the various hinges.*

- **Walker2d**: *The task is a reinforcement learning problem where an agent controls a 2-dimensional bipedal robot consisting of seven main body parts - a single torso at the top (with the two legs splitting after the torso), two thighs in the middle below the torso, two legs below the thighs, and two feet attached to the legs on which the entire body rests. The action space consists of 6 continuous values, each represents the torque applied at one hinge joints: {0: the right thigh rotor, 1: the right leg rotor, 2: the right foot rotor, 3: the left thigh rotor, 4: the left leg rotor, 5: the left foot rotor}. The goal is to make the robot walk forward (right) by applying torque to the six hinges that connect the seven body parts.*

# M  Discussion

## M.1  Limitation

One major limitation of our method lies in LLMs' illusion problem. Despite average performance improvement, LLMs may output unfaithful analysis under certain circumstances and poison specific training processes. When applied to real-world applications, these training trails may cause negative outcome. Therefore, how to verify the output of LLM agents and improve the reliability of our workflow worth future studies.

## M.2  Code of ethics

This study uses fully open-source or publicly available models and benchmarks, adhering to their respective licenses. All resources are properly cited in Sections 4.1. The selected benchmarks and models are well-established, representative, and free from bias or discrimination.

## M.3  Broader impacts

Our method holds significant potential to influence the broader domain of large language models (LLMs) and reinforcement learning (RL), a cross-research area that continues to attract substantial attention. Beyond the specific tasks demonstrated in our experiments, our approach is adaptable to a wider array of complex problems. Besides, its underlying design principles could inspire further research into leveraging LLMs to enhance various facets of RL algorithms—from policy representation and exploration strategies to reward shaping—ultimately fostering the development of more robust and intelligent AI systems.

