# OpenReview forum: "LLM-Explorer: A Plug-in Reinforcement Learning Policy Exploration Enhancement Driven by Large Language Models"
_NeurIPS.cc/2025/Conference — NeurIPS 2025 spotlight_

### Official Review · Reviewer_giQQ · 2025-06-20

**Clarity:** 4
**Significance:** 2
**Originality:** 3
**Rating:** 5
**Confidence:** 4

**Summary:**

This paper explores how to use LLM as a plug-in tool to enable better task-adaptive exploration in reinforcement learning instead of exploring with simple task-agnostic strategies like epsilon-greedy or Gaussian noise.
The LLM is prompted with task description, action sequence and reward history to analyze how to improve the exploration strategy and output a probability distribution for exploration in the end.
The effectiveness of the proposed method is validated on both Atari and Mujoco benchmarks, and carefully evaluated on different RL algorithms and LLM backbones.

**Questions:**

1. Do you use multiple historical episodes or just the latest episode as LLM input? Based on my understanding it's just the latest episode. What will happen if you feed in more episodes during the training process to the LLM? As a more thorough cover of explored regions may help the LLM better suggest what to explore next.
2. To confirm, does the LLM outputs only the probability distribution of exploration noise, instead of the state-conditioned action distribution? E.g., for discrete action, do you choose the highest-value action with probability (1-$\epsilon$), and only sample from the LLM-generated distribution with probability $\epsilon$?
3. Based on the figures in the paper, it seems that most training has not converged yet under the given training budget. I was wondering that how will your method and the baselines perform after convergence? This is motivated by two considerations: (1) The convergence results can help show if LLM-Explorer helps explore for better solution, or just accelerates the exploration process and leads to the same solution? (2) As LLM will takes considerable computations, I'm curious to see if the baseline methods can achieve similar performance if just given a larger training budget, so that they are compared under a similar total computational cost.
4. Do all the benchmarking tasks have the same initial state across episodes, given that you do not feed the state as an additional input to the LLM?
5. Atari and Mujoco are both very common benchmarks in RL, so I'm afraid that there may be a chance that the LLM already has a lot of prior knowledge about these tasks, and it just "tells" the policy what should be a better action to try next instead of truly proposing better exploration strategy based on the learning context. I would suggest to evaluate on some more challenging tasks that the LLM has less prior knowledge about, or discuss about the potential risk here in the paper.

**Ethical Concerns:**

["NO or VERY MINOR ethics concerns only"]

**Final Justification:**

The authors have addressed all my concerns during rebuttal with new experiments and detailed analysis, which better convinces me that their method is a general and effective approach to improve exploration in RL across a wide range of domains. So I will further raise my score to 5 and recommand acceptance of the paper.

**Limitations:**

Yes

**Quality:**

3

**Strengths And Weaknesses:**

Strengths:
1. The paper is well motivated and proposes an intuitive and effective solution to the exploration challenge;
2. The paper is well written and structured, and the methodology is clearly explained with sufficient technical details;
3. The evaluation is thorough, considering different action spaces, RL algorithms, LLM backbones, and carefully ablates some important design choices and hyperparameters in the method.

Weaknesses: I don't see an obvious weakness in the paper, but I do have some concerns about to what extent the proposed method is effective. See questions 3-5 below for more details.

---

> ### Author Rebuttal · Authors · 2025-07-31
>
> We sincerely thank the reviewer for the positive evaluation and insightful suggestions. Here we provide a point-to-point response with additional experimental results and discussions.
>
> **Q1. Multiple historical episodes input**
>
> Thank you for this point. Your understanding is correct: our current method updates the exploration strategy based on $M$ actions sampled uniformly from the single latest episode.
>
> Intuitively, providing data from more historical episodes could give the LLM a richer context for its analysis. To see what will actually happen, we test 2, 3, and 5 episodes, where we average 3 repeated trainings with different random seeds:
>
> |Task|DQN|DQN+LLM-Explorer (1 episode)||DQN+LLM-Explorer (2 episode)||DQN+LLM-Explorer (3 episode)||DQN+LLM-Explorer (5 episode)||
> |:-:|:-:|:-:|:-:|:-:|:-:|:-:|:-:|:-:|:-:|
> |||Input token (k)|Score|Input token (k)|Score|Input token (k)|Score|Input token (k)|Score|
> |Alien|245.26|248.73|268.44|277.73|273.19|306.73|275.18|364.73|267.49|
> |Freeway|5.25|220.12|20.64|249.32|21.26|278.52|21.44|336.92|20.73|
> |MsPacman|411.07|291.30|489.90|326.30|501.71|361.30|505.68|431.30|481.30|
>
> We find that performance does improve when using more historical data, but there is a bottleneck. The best performance is achieved when using data from approximately 3 episodes. When providing too many episodes, it may result in an input sequence that is overly long and complex, which can hinder the LLM's ability to effectively process the information. More inputs also lead to a proportional increase in computational cost.
>
> **Q2. Output format of exploration probability distribution**
>
> Thank you for this detailed question. Yes, your understanding is correct. The LLM outputs only the probability distribution for the exploration noise, not a complete state-conditioned action distribution.
> - For discrete actions, you are correct that our method integrates with an $\epsilon$-greedy framework. The agent chooses the highest-value action with probability $(1-\epsilon)$, and only with probability $\epsilon$ does it sample an action from the LLM-generated distribution.
> - For continuous actions, we generate a bias vector corresponding to the dimension of the action space, which is then added to the original symmetric exploration noise to create a biased, directional exploration distribution.
>
> We adopt this design for two main reasons. First, generating a full state-conditioned action distribution is highly complex, as states are often high-dimensional vectors or images that are difficult to obtain clean mathematical expressions. Second, by focusing only on the exploration distribution, our method produces a concise exploration signal that follows a form consistent with traditional methods. This ensures that LLM-Explorer functions as a simple plug-in module, making it easily compatible with a wide range of existing RL algorithms.
>
> **Q3. Convergence and computational cost**
>
> Thanks for this question.
>
> **First**, our primary objective with LLM-Explorer is to improve the efficiency of the training process, enabling the agent to achieve better performance within a limited number of environment steps. This focus on sample efficiency is a standard paradigm in existing works [1-3], such as the Atari 100k challenge, and is the main goal of our work.
>
> **Second**, we acknowledge the reviewer's point that full convergence can reveal different aspects of an algorithm's performance. As established by prior work [4], reaching full convergence on these tasks can require tens of millions of steps. Due to time constraints, we conduct a new experiment where we train agents for 1 million steps, a point at which the learning curves show a clear trend toward convergence. To address the computational cost concern and ensure a fair comparison, we allow the baseline DQN algorithm to train for more steps, matching its total wall-clock time to the extra LLM consumption in LLM-Explorer. For each task, we average 3 repeated trainings with different random seeds:
>
> |Task|DQN+LLM-Explorer||DQN||
> |:-:|:-:|:-:|:-:|:-:|
> ||Steps (M)|Score|Steps (M)|Score|
> |Alien|1.00|458.70|1.11|431.67|
> |Freeway|1.00|36.23|1.03|31.87|
> |MsPacman|1.00|1412.34|1.12|1409.32|
>
> The results indicate that the final performance of LLM-Explorer is similar to, and in some environments slightly better than, the baseline. This aligns with our design goal. It demonstrates that our method primarily enhances exploration efficiency during the critical early and middle phases of training, while also ensuring that the agent reaches a high-quality final solution that does not sacrifice convergence performance.
>
> [1] Model-based reinforcement learning for Atari. ICLR 2020.\
> [2] Data-Efficient Reinforcement Learning with Self-Predictive Representations. ICLR 2021.\
> [3] Image augmentation is all you need: Regularizing deep reinforcement learning from pixels. ICLR 2022.\
> [4] Bigger, better, faster: Human-level atari with human-level efficiency. ICML 2023.
>
> **Q4-5. State awareness and LLM’s prior knowledge about tasks**
>
> Thanks for this critical point. The initial state for each episode in the benchmark environments is randomly initialized as the default settings according to the gym package. Actually, our state-agnostic design is a deliberate choice. Here, we provide more discussions to better explain and improve our approach.
>
> **First**, omitting state information is a trade-off between performance and efficiency. States in RL are often high-dimensional vectors or images, which are computationally expensive to process. As our experiments demonstrate, analyzing the previous action-reward trajectory, in conjunction with the task definition, is sufficient to guide exploration effectively.
>
> **Second**, you are right that the reason why the method can operate without the state information may be the dependence on the LLM's prior knowledge of the tasks. To explore this, we conduct a simple test:
> 1. We first state the required format of the task description and manually craft a task description for the 'Alien' game from the Atari.
> 2. We then prompt GPT-4o mini to generate task descriptions for other environments with the one-shot example.
>
> We find that the LLM can generate accurate descriptions from the game names alone, indicating prior knowledge of the well-known tasks. To test the framework's ability independent of this prior knowledge, we select three less common Atari games (outside the 26 core games [1]). For these games, we confirm that the LLM is unable to generate correct task descriptions from the name alone, suggesting a lack of specific prior knowledge.
>
> We then manually provide detailed task descriptions for these games and train agents. For each task, we average 3 repeated trainings with different random seeds:
>
> |Task|DQN|DQN+LLM-Explorer|
> |:-:|:-:|:-:|
> |Berzerk|142.38|171.23|
> |Centipede|2340.34|3160.48|
> |Gravitar|366.71|425.76|
>
> The results show performance improvement, demonstrating that our method is not solely dependent on the LLM's prior knowledge. Instead, it can analyze the provided task description and the action-reward data to produce an effective exploration strategy.
>
> **Third**, we acknowledge that it is difficult to completely rule out the influence of all implicit knowledge within the LLM. In scenarios that are entirely novel, the performance of our state-agnostic method may be limited. For these cases, incorporating state information via a Vision-Language Model (VLM) is a possible solution. Specifically, we design an extended experiment:
> - We select Qwen2.5-VL-32B as the VLM. At the end of every $K$ episodes, we input the final state image from the last episode, along with the corresponding action sequence and episode reward, to the VLM. The final state image provides critical situational context regarding the agent's success or failure in that episode.
> - For a controlled comparison, we also run an experiment using the text-only counterpart, Qwen2.5-32B. This model follows our original state-agnostic design.
>
> For each task, we average 3 repeated trainings with different random seeds:
>
> |Task|DQN|DQN+LLM-Explorer (Qwen2.5-32B)|||DQN+LLM-Explorer (Qwen2.5-VL-32B)|||
> |:-:|:-:|:-:|:-:|:-:|:-:|:-:|:-:|
> ||Score|Score|Cost ($)|Time (h)|Score|Cost ($)|Time (h)|
> |Alien|245.46|262.84|0.22|1.85|271.67|0.91|2.39|
> |Freeway|5.25|17.61|0.18|5.15|19.23|0.72|5.61|
> |MsPacman|411.07|480.15|0.25|1.87|497.21|1.04|2.50|
>
> Comparing the outcomes, the VLM-based approach shows an improvement. However, this comes at the cost of higher computational overhead due to the VLM's complexity.
>
> In general, this highlights a trade-off. Our original state-agnostic design is intentionally lightweight, which is a practical and efficient solution for general use. In contrast, for users who wish to maximize performance without budget limitations, the VLM-based extension can be beneficial. We will also add a more detailed discussion around the potential risk of this design to the limitations section of our paper.
>
> [1] Model-based reinforcement learning for Atari. ICLR 2020.
>
> Thanks again for your valuable suggestions! We will include the above discussions and improvements into our final version. Please let us know if you have any further concerns.

---

> > ### Comment · Reviewer_giQQ · 2025-08-04
> >
> > Dear authors,
> >
> > Thanks for your detailed reply and extended experiments, which address most of my questions. I have two follow up questions:
> > 1. For the new results, could you please also report the standard error across seeds?
> > 2. I understand that it is hard to include state information in the query to LLM, but I still don't quite get how the LLM works without initial state information. Executing the same sequence of actions from different initial states will lead to different results, and as the initial state is not fed as an input to the LLM, how can it tell if the current action sequence is better or not and how to improve the action distribution correspondingly?

---

> > > ### Author Response · Authors · 2025-08-04
> > >
> > > Dear reviewer,
> > >
> > > Thanks again for your reviews. We are delight that our responses have addressed your major concerns. Here are discussions on the follow up question.
> > >
> > > **Q1. Standard error across seeds**
> > >
> > > Here we show [std] across seeds for our new results:
> > >
> > > **Multiple episodes input experiment**
> > > |Task|DQN|DQN+LLM-Explorer (1 episode)||DQN+LLM-Explorer (2 episode)||DQN+LLM-Explorer (3 episode)||DQN+LLM-Explorer (5 episode)||
> > > |:-:|:-:|:-:|:-:|:-:|:-:|:-:|:-:|:-:|:-:|
> > > |||Input token (k)|Score|Input token (k)|Score|Input token (k)|Score|Input token (k)|Score|
> > > |Alien|245.26 [1.38]|248.73 [0.67]|268.44 [11.27]|277.73 [0.74]|273.19 [12.21]|306.73 [0.75]|275.18 [11.89]|364.73 [0.80]|267.49 [10.94]|
> > > |Freeway|5.25 [3.27]|220.12 [0.02]|20.64 [0.25]|249.32 [0.03]|21.26 [0.27]|278.52 [0.03]|21.44 [0.34]|336.92 [0.05]|20.73 [0.24]|
> > > |MsPacman|411.07 [10.55]|291.30 [0.82]|489.90 [30.55]|326.30 [1.27]|501.71 [35.22]|361.30 [0.88]|505.68 [34.78]|431.30 [1.01]|481.30 [27.60]|
> > >
> > > **1M steps toward convergence experiment**
> > > |Task|DQN+LLM-Explorer||DQN||
> > > |:-:|:-:|:-:|:-:|:-:|
> > > ||Steps (M)|Score|Steps (M)|Score|
> > > |Alien|1.00|458.70 [19.10]|1.11|431.67 [4.61]|
> > > |Freeway|1.00|36.23 [0.41]|1.03|31.87 [2.28]|
> > > |MsPacman|1.00|1412.34 [41.29]|1.12|1409.32 [27.06]|
> > >
> > > **Atari tasks with less prior knowledge experiment**
> > > |Task|DQN|DQN+LLM-Explorer|
> > > |:-:|:-:|:-:|
> > > |Berzerk|142.38 [10.69]|171.23 [25.73]|
> > > |Centipede|2340.34 [58.27]|3160.48 [203.12]|
> > > |Gravitar|366.71 [18.39]|425.76 [42.08]|
> > >
> > > **VLM with final state input experiment**
> > > |Task|DQN|DQN+LLM-Explorer (Qwen2.5-32B)|||DQN+LLM-Explorer (Qwen2.5-VL-32B)|||
> > > |:-:|:-:|:-:|:-:|:-:|:-:|:-:|:-:|
> > > ||Score|Score|Cost ($)|Time (h)|Score|Cost ($)|Time (h)|
> > > |Alien|245.46 [1.38]|262.84 [9.49]|0.22 [0.00]|1.85 [0.01]|271.67 [13.52]|0.91 [0.01]|2.39 [0.03]|
> > > |Freeway|5.25 [3.27]|17.61 [0.17]|0.18 [0.00]|5.15 [0.05]|19.23 [0.22]|0.72 [0.00]|5.61 [0.13]|
> > > |MsPacman|411.07 [10.55]|480.15 [27.60]|0.25 [0.00]|1.87 [0.02]|497.21 [35.28]|1.04 [0.01]|2.50 [0.05]|
> > >
> > >
> > > **Q2. Random initial states of environments**
> > >
> > > We can understand the intuition you mentioned. Let's imagine an extreme task: "get to the center of the room." If in the first episode, the agent is randomly initialized on the left side, then "moving right" actions are beneficial. But if the next episode starts on the right, the LLM, being unaware of this new initial state, would still favor exploring "move right" based on the last episode's experience.
> > >
> > > However, most tasks, including those in our benchmarks, are not as extreme as this hypothetical example. Our state-agnostic approach works based on two main factors:
> > > - **Structure similarity of initial states.**: While the initial positions of elements like the agent, obstacles (e.g., cars in Freeway), and enemies (e.g., ghosts in MsPacman) are randomized, their overall structure and relationships between these elements are generally consistent across episodes.
> > > - **Dynamic similarity of agent-environment interactions.** Most tasks involve long and complex processes, not just simple, few-step action sequences. The underlying dynamics of how the agent interacts with the environment are largely the same regardless of the specific starting positions.
> > >
> > > Thereby, the effective action distribution tends to have strong similarities across different episodes of the same task. LLM-Explorer can analyze these general patterns based on the task description and action-reward history to improve the exploration distribution.
> > > - **For instance**, as shown in our case study of Freeway (Figure 4), the agent may start at different positions on the bottom of road and car patterns may vary, but to achieve the goal of "reaching the top of the road", the action "moving up" is almost always beneficial, while "no-ops" is generally ineffective. The LLM can correctly infer this state-agnostic strategic bias.
> > >
> > > That said, if a task truly matches the extremity of the hypothetical example we imagined above, our current LLM-Explorer framework would likely perform poorly. For such cases, the framework can be extended to incorporate a VLM, which would receive the initial state images as an additional input alongside the task descriptions.
> > > - **For example**, if the Freeway task were modified so the goal is to "reach the opposite side" and the agent could start at either the top or the bottom, the VLM would need the initial state to determine whether "moving up" or "moving down" should be prioritized.
> > >
> > > In general, for most normal tasks, the current state-agnostic design is a good choice considering computational cost, while for some extreme situations, a VLM version may be required to incorporate initial state information.
> > >
> > > We will try to find or construct some kind of tasks with extremely different initial states, such as a modified bi-directional Freeway, to validate our analyses and add a detailed discussion to the final version.
> > > \
> > > \
> > > \
> > > Thanks again for helping us improve our work. Please let us know if you have any further concerns.

---

> > > > ### Comment · Reviewer_giQQ · 2025-08-04
> > > >
> > > > Thanks for your reply! I have no further questions, and am happy to raise my score to 5 given that the authors have addressed all my concerns with new experimental results and detailed analysis.

---

> > > > > ### Author Response · Authors · 2025-08-04
> > > > >
> > > > > Thanks again for the suggestions and discussions! We are glad to improve the final version of our paper accordingly.

---

### Official Review · Reviewer_78Fn · 2025-06-24

**Clarity:** 4
**Significance:** 2
**Originality:** 3
**Rating:** 5
**Confidence:** 4

**Summary:**

The manuscript proposes LLM-Explorer, a plug-in module that leverages large language models to enhance policy exploration in reinforcement learning. Unlike traditional exploration strategies that rely on preset stochastic processes, LLM-Explorer adaptively generates task-specific exploration strategies by analyzing the agent’s learning trajectories and dynamically adjusting the probability distribution for future actions. The framework employs two LLMs that interact via natural language prompts to summarize the agent’s learning status and suggest exploration strategies, which are then converted into actionable probability distributions for RL agents. The design is compatible with a wide range of RL algorithms and has been validated through extensive experiments on discrete and continuous action space benchmarks, demonstrating notable performance improvements.

**Questions:**

1. Table 4 indicates that a temperature parameter of 1.0 is used for all LLMs, which introduces high output variance. Would lowering the temperature improve the stability and reproducibility of the method?

2. Since LLM-Explorer updates the exploration strategy at every interval, is it possible to reduce the frequency of LLM interactions without significantly impacting performance? Are there heuristic or adaptive mechanisms to trigger LLM updates only when necessary?

3. The method relies on frequent interactions with large LLMs, which can be resource-intensive. Have the authors considered or experimented with smaller, open-source, or fine-tuned LLMs to reduce computational costs? What is the minimal LLM size that still yields notable performance improvements?

**Ethical Concerns:**

["NO or VERY MINOR ethics concerns only"]

**Final Justification:**

This manuscript presents LLM-Explorer, a novel and modular approach that leverages large language models to enhance policy exploration in reinforcement learning. The authors provide extensive experimental evidence across diverse RL benchmarks, demonstrating consistent and significant performance improvements over baseline methods. The framework’s plug-in design ensures broad compatibility and practical usability, while the comprehensive ablation studies and additional analyses included in the authors’ response further validate the method’s effectiveness and flexibility. Although some open questions remain regarding the influence of LLM prior knowledge and computational trade-offs, the authors’ willingness to address these issues and their proposed directions for future work are commendable. Overall, the manuscript offers a valuable contribution to the field and is likely to inspire further research. I therefore recommend acceptance.

**Limitations:**

Yes

**Quality:**

3

**Strengths And Weaknesses:**

## Strengths
1. The manuscript tries to address a novel and interesting problem by using LLMs to guide exploration in RL, instead of building general RL agents with LLMs, which is a prevalent focus in current research. If the exploration strategy is efficient enough, then the former has stronger usability.

2. The proposed method is simple, modular, and the writing is easy to understand.

3. The plug-in design of LLM-Explorer ensures broad compatibility, supporting a variety of RL algorithms across both discrete and continuous action spaces.

4. The experimental evaluation is comprehensive, covering multiple base RL algorithms and benchmark environments, and demonstrates significant performance gains.

## Weaknesses

1. The manuscript lacks a detailed analysis of computational overhead, including the number of tokens consumed per training run and the additional wall-clock time incurred by frequent LLM interactions. High resource consumption could limit the practical applicability of the proposed method.

2. The method currently uses only action-reward trajectories as input to the LLMs. The effectiveness of the methods demonstrated in the experimental results may largely depend on LLM's knowledge related to these tasks, and the agent's learning process can be reflected solely through action sequences. For more complex tasks, omitting state information may restrict the method’s generality and effectiveness.

3. Only three random seeds are used for each experiment. Given the inherent stochasticity in both RL training and LLM outputs, this is insufficient to robustly assess the method’s stability and reproducibility.

---

> ### Author Rebuttal · Authors · 2025-07-31
>
> We sincerely thank the reviewer for the positive evaluation and insightful suggestions. Here we provide a point-to-point response with additional experimental results and discussions.
>
> **W1. Computational overhead**
>
> We agree that computational time and cost is crucial. Here, we report the time and API cost for 500k-step training. For each task, we average 3 repeated trainings with different random seeds:
>
> |Environment|Task|Input token (k)|Output token (k)|Cost ($)|Time (h)|
> |:-:|:-:|:-:|:-:|:-:|:-:|
> |Atari|Alien|1243.65|897.95|0.73|9.23|
> ||Amidar|1563.54|927.00|0.79|9.37|
> ||BankHeist|1514.98|1101.00|0.89|9.42|
> ||Breakout|3821.06|2634.00|2.15|11.02|
> ||ChopperCommand|1340.64|684.00|0.61|9.18|
> ||CrazyClimber|413.23|208.00|0.19|8.59|
> ||Freeway|366.87|231.25|0.19|8.58|
> ||Hero|1191.81|686.00|0.59|9.12|
> ||Jamesbond|2133.88|1011.00|0.93|9.64|
> ||Krull|805.92|438.00|0.38|8.85|
> ||MsPacman|1456.50|1006.10|0.82|9.36|
> ||Pong|260.83|312.00|0.23|8.57|
> ||Qbert|2263.39|1626.00|1.32|9.95|
> ||Seaquest|1389.57|768.00|0.67|9.23|
> ||UpNDown|1447.75|1011.00|0.82|9.36|
> |MuJoCo|HalfCheetah|978.16|524.98|0.46|8.96|
> ||Hopper|5693.25|3611.96|3.02|12.21|
> ||Humanoid|25387.59|7477.76|8.29|22.03|
> ||HumanoidStandup|2061.12|610.47|0.68|9.45|
> ||Walker2d|4422.74|2475.20|2.15|11.21|
> ||**Average**|**2987.82**|**1412.08**|**1.30**|**10.17**|
>
> On average, each run consumes $1.3 USD in API calls and takes about 10 hours. We consider this to be an acceptable and reasonable overhead.
>
> **W2. State awareness and LLM's prior knowledge about tasks**
>
> Thanks for this critical point. We provide more discussions to better explain and improve our approach.
>
> **First**, omitting state information is a trade-off between performance and efficiency. States in RL are often high-dimensional vectors or images, which are computationally expensive to process. As our experiments demonstrate, analyzing the previous action-reward trajectory, in conjunction with the task definition, is sufficient to guide exploration effectively.
>
> **Second**, you are right that the method may depend on the LLM's prior knowledge of the tasks. To explore this, we conduct a simple test:
> 1. We first state the required format of the task description and manually craft a task description for the 'Alien' game from the Atari.
> 2. We then prompt GPT-4o mini to generate task descriptions for other environments with the one-shot example.
>
> We find that the LLM can generate accurate descriptions from the game names alone, indicating prior knowledge of the well-known tasks. To test the framework's ability independent of this prior knowledge, we select three less common Atari games (outside the 26 core games [1]). For these games, we confirm that the LLM is unable to generate correct task descriptions from the name alone, suggesting a lack of specific prior knowledge.
>
> We then manually provide detailed task descriptions for these games and train agents. For each task, we average 3 repeated trainings with different random seeds:
>
> |Task|DQN|DQN+LLM-Explorer|
> |:-:|:-:|:-:|
> |Berzerk|142.38|171.23|
> |Centipede|2340.34|3160.48|
> |Gravitar|366.71|425.76|
>
> The results show performance improvement, demonstrating that our method is not solely dependent on the LLM's prior knowledge. Instead, it can analyze the provided task description and the action-reward data to produce an effective exploration strategy.
>
> **Third**, we acknowledge that it is difficult to completely rule out the influence of all implicit knowledge within the LLM. In scenarios that are entirely novel, the performance of our state-agnostic method may be limited. For these cases, incorporating state information via a Vision-Language Model (VLM) is a possible solution. Specifically, we design an extended experiment:
> - We select Qwen2.5-VL-32B as the VLM. At the end of every $K$ episodes, we input the final state image from the last episode, along with the corresponding action sequence and episode reward, to the VLM. The final state image provides critical situational context regarding the agent's success or failure in that episode.
> - For a controlled comparison, we also run an experiment using the text-only counterpart, Qwen2.5-32B. This model follows our original state-agnostic design.
>
> For each task, we average 3 repeated trainings with different random seeds:
>
> |Task|DQN|DQN+LLM-Explorer (Qwen2.5-32B)|||DQN+LLM-Explorer (Qwen2.5-VL-32B)|||
> |:-:|:-:|:-:|:-:|:-:|:-:|:-:|:-:|
> ||Score|Score|Cost ($)|Time (h)|Score|Cost ($)|Time (h)|
> |Alien|245.46|262.84|0.22|1.85|271.67|0.91|2.39|
> |Freeway|5.25|17.61|0.18|5.15|19.23|0.72|5.61|
> |MsPacman|411.07|480.15|0.25|1.87|497.21|1.04|2.50|
>
> Comparing the outcomes, the VLM-based approach shows an improvement. However, this comes at the cost of higher computational overhead due to the VLM's complexity.
>
> In general, this highlights a trade-off. Our original state-agnostic design is intentionally lightweight, which is a practical and efficient solution for general use. In contrast, for users who wish to maximize performance without budget limitations, the VLM-based extension can be beneficial.
>
> [1] Model-based reinforcement learning for Atari. ICLR 2020.
>
> **W3. More random seeds**
>
> Here, we increase the number of random seeds from 3 to 5 for all of our Atari experiments:
>
> ||DQN||DQN+LLM-Explorer||Improvement (%)|
> |:-:|:-:|:-:|:-:|:-:|:-:|
> ||Score [std]|Human-norm score (%)|Score [std]|Human-norm score (%)||
> |Alien|245.55 [0.77]|0.26|269.25 [6.30]|0.60|130.77|
> |Amidar|22.96 [0.89]|1.00|28.42 [2.40]|1.32|32.00|
> |BankHeist|18.43 [1.22]|0.57|19.32 [1.12]|0.69|21.05|
> |Breakout|2.74 [0.10]|3.62|2.86 [0.15]|4.01|10.77|
> |ChopperCommand|840.84 [8.14]|0.45|868.85 [20.46]|0.88|95.56|
> |CrazyClimber|16952.18 [356.98]|24.64|17463.78 [694.11]|26.68|8.28|
> |Freeway|5.16 [2.21]|17.43|20.55 [0.17]|69.43|298.34|
> |Hero|1448.19 [212.65]|1.41|2586.98 [144.07]|5.23|270.92|
> |Jamesbond|59.03 [7.64]|10.97|76.97 [1.60]|17.52|59.71|
> |Krull|2915.30 [23.79]|123.40|2950.67 [78.36]|126.71|2.68|
> |MsPacman|413.68 [6.62]|1.60|497.48 [19.19]|2.86|78.75|
> |Pong|-15.80 [0.21]|13.87|-14.31 [0.41]|18.10|30.50|
> |Qbert|304.85 [7.88]|1.06|300.58 [8.96]|1.03|-2.83|
> |Seaquest|201.03 [2.16]|3.17|193.55 [10.13]|2.99|-5.68|
> |UpNDown|1335.94 [110.10]|7.19|1459.16 [95.43]|8.30|15.44|
> |**Average**|**1639.27**|**15.03**|**1781.61**|**20.41**|**35.80**|
>
> The results from the 5-seed runs are consistent with our original findings. Due to the time limit, we will add more random seeds for other experiments and update the results into the final version of our paper.
>
> **Q1. Temperature and stability**
>
> Thank you for noticing this detail. Actually, we have investigated the temperature parameter in our research process. With various temperatures, we average 3 repeated trainings with different random seeds for each task:
>
> |Task|DQN|LLM-Explorer (GPT-4o temperature=0.5)|LLM-Explorer (GPT-4o temperature=1.0)|LLM-Explorer (GPT-4o temperature=2.0)|
> |:-:|:-:|:-:|:-:|:-:|
> |Alien|245.46|243.89|249.19|235.43|
> |Freeway|5.25|15.23|19.95|4.74|
> |MsPacman|411.07|412.97|415.60|394.19|
>
> While a lower temperature reduces output variance, we find that it also leads to less diverse exploration strategies, resulting in limited performance gains. Therefore, we turn to the current setting of 1.0. But further increasing the temperature leads to a sharp performance decline. This suggests that excessively high temperatures introduce meaningless randomness rather than beneficial, flexible exploration. This also aligns with general recommendations from providers like OpenAI.
>
> **Q2. Adaptive update interval**
>
> This is a very insightful suggestion. Our existing results have shown that simply reducing the update frequency leads to a decline in performance (Table 3). Following your suggestion, we design a simple heuristic mechanism to trigger LLM updates more adaptively. After each episode, we calculate the average score improvement over the last 5 episodes ($g_1$) and the last 10 episodes ($g_2$). If $g_1$ is lower than $g_2$ by more than a given threshold $G$, we infer that the current exploration strategy may be losing effectiveness and trigger an LLM update. For each task, we average 3 repeated trainings with different random seeds:
>
> |Task|DQN|DQN+LLM-Explorer (fix K=1)||DQN+LLM-Explorer (fix K=2)||DQN+LLM-Explorer (adaptive G=10%)||DQN+LLM-Explorer (adaptive G=30%)||
> |:-:|:-:|:-:|:-:|:-:|:-:|:-:|:-:|:-:|:-:|
> |||Avg. update interval|Score|Avg. update interval|Score|Avg. update interval|Score|Avg. update interval|Score|
> |Alien|245.26|1.00|268.44|2.00|254.02|1.51|265.19|1.86|261.43|
> |Freeway|5.25|1.00|20.64|2.00|19.16|1.64|20.51|1.98|19.82|
> |MsPacman|411.07|1.00|489.90|2.00|454.80|1.47|486.29|1.74|479.18|
>
> The results show reduced average update frequency. Also, the resulting performance degradation is less severe than a fixed update interval change. We believe that more sophisticated adaptive triggers are valuable for future research to improve the performance-cost balance.
>
> **Q3. Smaller LLMs**
>
> It is interesting to explore the minimal capable LLM size. We test multiple Qwen2.5 models with varying sizes. For each task, we average 3 repeated trainings with different random seeds:
>
> |Task|DQN|LLM-Explorer (Qwen2.5-32B)|LLM-Explorer (Qwen2.5-14B)|LLM-Explorer (Qwen2.5-7B)|LLM-Explorer (Qwen2.5-3B)|
> |:-:|:-:|:-:|:-:|:-:|:-:|
> |Alien|245.46|262.84|251.41|239.18|231.98|
> |Freeway|5.25|17.61|9.11|7.79|5.01|
> |MsPacman|411.07|480.15|435.97|407.48|383.17|
>
> The performance degrades with smaller model sizes, and $\leq$7B parameters generally fail to work. Also, as you mentioned, fine-tuning a specialized LLM, for example, distilling the exploration strategy generation from a large LLM into a smaller one, would further reduce the minimum LLM size and save more computational cost.
>
> Thanks again for your valuable suggestions! We will include the above discussions and improvements into our final version. Please let us know if you have any further concerns.

---

> > ### Comment · Reviewer_78Fn · 2025-08-01
> >
> > Thank you for your thorough responses, as well as for the substantial additional experimental results and analyses. Your clarifications and extended experiments have addressed the majority of my concerns and further substantiated both the effectiveness and the flexibility of the proposed LLM-Explorer framework. The inclusion of adaptive update mechanisms, a more nuanced discussion of computational overhead, and experiments involving vision-language models significantly strengthens the manuscript and highlights its extensibility for a variety of use cases.
> >
> > That said, I would like to reiterate and slightly expand upon a key point regarding the influence of LLMs’ prior knowledge. While your experiments with less common Atari games are appreciated, it remains challenging to fully disentangle the effect of implicit knowledge embedded within large language models. As my own tests with GPT-4o mini suggest, even for these games, the model can often generate plausible—if not entirely accurate—descriptions. Furthermore, the underlying task logic in many Atari games shares considerable structural similarity, which may limit the conclusiveness of these tests in isolating the method’s reliance on prior knowledge. Designing or employing tasks that are fundamentally different from the existing benchmarks, or at least less correlated with the standard RL benchmarks, would provide a more rigorous evaluation of the generality and adaptability of LLM-Explorer in truly novel environments. I encourage the authors to consider this direction for future work, as it would greatly strengthen the empirical claims of the paper.
> >
> > Regarding computational cost, while you claim that the additional expense is reasonable for the current scale of experiments, the trade-off between performance gains and resource requirements remains an open question, especially for larger-scale applications. It would be valuable for the final version to include a more explicit discussion of these trade-offs, as well as the potential for further optimization through model distillation or other efficiency-focused approaches.
> >
> > Overall, I am pleased with the authors’ comprehensive responses and willingness to incorporate additional analyses into the final version. The empirical depth and modular design of LLM-Explorer offer clear value to the reinforcement learning community, and I believe the work opens up promising avenues for future research. I am thus inclined to recommend acceptance.

---

> > > ### Author Response · Authors · 2025-08-01
> > >
> > > Thanks for your timely reply!
> > >
> > > It is true that completely isolating the method’s reliance on prior knowledge is a challenging problem, and our additional results serve as a preliminary step in this direction. In future work, we are willing to test with more fundamentally different benchmarks or consider some more sophisticated experimental designs to exclude the impact of prior knowledge. These would vastly enhance the understanding of the generality and adaptability of LLM-Explorer.
> > >
> > > Regarding the trade-off between performance gains and resource requirements, it is always beneficial to further enhance efficiency. Notably, as you pointed out, larger-scale and more complex applications may require more resource consumption, even the enrollment of large VLMs, where approaches to improve efficiency are critical. We will include discussions about the performance-cost trade-offs in the final version, as well as potential optimization directions, such as smaller model distillation and adaptive update mechanisms that we have discussed.
> > >
> > > Thanks again for your insightful suggestions and targeted feedback. We will polish the final version of our paper accordingly.

---

### Official Review · Reviewer_65Q1 · 2025-07-02

**Clarity:** 2
**Significance:** 2
**Originality:** 3
**Rating:** 4
**Confidence:** 4

**Summary:**

This paper presents LLM-Explorer, a novel plug-in module that enhances policy exploration in reinforcement learning (RL) by leveraging Large Language Models (LLMs). The work addresses the key limitations of traditional exploration strategies, which use preset stochastic processes that are rigid and non-adaptive to specific tasks or an agent's real-time learning status. LLM-Explorer's methodology is a two-stage process. It periodically samples an agent's action-reward trajectory, which a primary LLM analyzes to summarize the learning status and provide strategic recommendations. A second LLM uses this summary to generate a tailored probability distribution for the next phase of exploration. This design is versatile, supporting both discrete and continuous action spaces and integrating seamlessly with popular RL algorithms like DQN, DDPG, and TD3. Through extensive experiments on the Atari and MuJoCo benchmarks, LLM-Explorer demonstrates significant performance improvements, achieving up to a 37.27% average increase in human-normalized scores on Atari tasks. The paper further validates its robustness and compatibility across various RL algorithms and different LLMs.

**Questions:**

1. **On Failure Modes and Applicability Limits:**  While the average performance gains are significant, results slightly declined in games like 'Qbert' and 'Seaquest'. Could you elaborate on the potential failure modes of LLM-Explorer? What characteristics of a task, such as a complex reward landscape or an ambiguous task description, might cause the LLM to generate suboptimal exploration strategies?
2. **On the Counter-intuitive LLM Performance:**  The finding that GPT-4o mini outperforms the more powerful GPT-4o is fascinating. Could you expand on your hypothesis that larger models "greedily fit specific actions"? Have you experimented with prompt engineering (e.g., instructing the LLM to be more creative) or adjusting temperature to see if the performance of larger models could be improved?
3. **On Reliability and Safeguards:** You rightly acknowledge the risk of LLM "hallucinations" negatively impacting training. Have you considered implementing practical safeguards to mitigate this? For example, could a "sanity check" be applied to the LLM's output distribution to prevent extreme policies, or could a monitoring system revert to a default strategy if rewards drop sharply to ensure stability?

**Ethical Concerns:**

["NO or VERY MINOR ethics concerns only"]

**Final Justification:**

The response from the authors have addressed my concerns, and there is not misunderstaning about the technical contributions. Although the limitations of the proposed method cannot be eliminated, the author makes deeper analysis of the technical mechanism, failure modes and safeguards. So I raise my score to boardline accept.

**Limitations:**

Limitations are fairly discussed.

**Quality:**

2

**Strengths And Weaknesses:**

**strengths**
1. **Novel Conceptual Framework for Exploration:** The paper's primary strength is its innovative approach to solving a fundamental problem in reinforcement learning—the rigidity of exploration strategies. Instead of relying on preset stochastic processes like ε-greedy or Gaussian noise, the authors introduce a new paradigm that leverages the analytical and reasoning capabilities of Large Language Models (LLMs). By dynamically generating exploration policies based on an analysis of the agent's action-reward trajectories, the method allows for exploration strategies that are adaptive to both the specific task and the agent's real-time learning progress. This is a significant conceptual shift from traditional methods.
2. **Extensive Empirical Validation and Versatility:** A second major strength is the method's rigorously demonstrated effectiveness and versatility. The authors validate LLM-Explorer across diverse and challenging benchmarks (Atari for discrete actions and MuJoCo for continuous actions), showing substantial performance gains (up to 37.27% average improvement on Atari). Furthermore, the work proves the method's practicality by designing it as a plug-in module and confirming its compatibility with a wide range of established RL algorithms, including the DQN family, DDPG, and TD3. This comprehensive evaluation strongly substantiates the authors' claims and highlights the method's broad applicability.

**weaknesses**
1. **Heavy reliance on prompt engineering:** Performance depends critically on carefully crafted, environment-specific prompts. Generalizability to broader domains is uncertain.
2. **Insufficient failure analysis:** Limited discussion of when and why the method fails or underperforms compared to traditional exploration methods.
3. **Weak baselines:** Only compares against NoisyNet and RND, lacking evaluation against recent state-of-the-art adaptive exploration techniques.

---

> ### Author Rebuttal · Authors · 2025-07-31
>
> We sincerely thank the reviewer for the insightful suggestions. Here we provide a point-to-point response with additional experimental results and discussions.
>
> **Q1. Automatic task description prompting**
>
> We agree that this is an important point.
>
> To improve our framework, we design to employ an additional LLM to automatically generate the task descriptions:
> 1. We first state the required format of the task description (line 120) and manually craft a task description for the 'Alien' game (Appendix G).
> 2. We then prompt GPT-4o mini to generate task descriptions for other environments with the one-shot example.
> ```
> Prompt: You are writing a task description for {TaskName} to support the subsequent task solving. The required format is {TaskDescriptionFormat}. Here is an example for the Alien game from the Atari benchmark: {TaskDescriptionExample}. Please strictly follow the format and cover all details of actions, rewards, end conditions, and goals.
> ```
> 3. We replace the original, hand-written task descriptions with the automatically generated ones and re-run the experiments. For each task, we average 3 repeated trainings with different random seeds:
>
> |Environment|Task|DQN|DQN+LLM-Explorer (manual)|DQN+LLM-Explorer (auto)|
> |:-:|:-:|:-:|:-:|:-:|
> |Atari|Alien|245.46|268.44|--|
> ||Amidar|22.34|26.75|25.16|
> ||BankHeist|18.64|19.51|19.60|
> ||Breakout|2.67|2.74|2.71|
> ||ChopperCommand|840.63|868.33|873.98|
> ||CrazyClimber|17070.76|17694.35|17731.42|
> ||Freeway|5.25|20.64|19.41|
> ||Hero|1439.70|2689.62|2513.76|
> ||Jamesbond|60.48|77.35|78.13|
> ||Krull|2933.05|3009.12|3013.67|
> ||MsPacman|411.07|489.90|501.13|
> ||Pong|-15.71|-14.13|-14.07|
> ||Qbert|306.07|301.97|303.53|
> ||Seaquest|201.58|196.15|202.03|
> ||UpNDown|1370.99|1489.54|1476.72|
> |MuJoCo|HalfCheetah|1342.75|1363.51|1358.64|
> ||Hopper|293.50|466.80|447.23|
> ||Humanoid|562.18|568.16|569.57|
> ||HumanoidStandup|78272.11|107257.07|106735.86|
> ||Walker2d|376.27|482.57|477.86|
> ||**Average**|**5553.39**|**7211.05**|**7175.60**|
>
> As the results show, LLM-Explorer with the auto-generated task descriptions performs similarly to the original one. This means that generating task descriptions can be automated without a significant drop in performance, which strengthens the scalability of our method to a broader range of tasks.
>
> **Q2. Failure mode and safeguards**
>
> It is true that LLM-Explorer is not universally effective and, as you pointed out, its performance is limited in certain tasks like Qbert and Seaquest.
>
> According to our experiences and suggestions from Reviewer NcaS, we consider that one possible reason is that the LLM outputs extremely skewed distributions in some cases, which harm the training.
> To diagnose, we calculate the KL divergence of each distribution generated during training to a uniform distribution and examine the deviation between the maximum and average values.
>
> |Task|Action dimension|$Max[KL(p,U)]$|$\frac{Max[KL(p,U)]-Mean[KL(p,U)]}{std[KL(p,U)]}$|
> |:-:|:-:|:-:|:-:|
> |Alien|18|0.773|8.298|
> |Amidar|10|1.205|5.037|
> |BankHeist|18|2.082|12.477|
> |Breakout|4|0.436|5.985|
> |ChopperCommand|18|1.969|13.319|
> |CrazyClimber|9|1.034|10.104|
> |Freeway|3|0.58|4.458|
> |Hero|18|0.847|7.385|
> |Jamesbond|18|1.085|8.552|
> |Krull|18|1.969|11.585|
> |MsPacman|9|1.182|4.751|
> |Pong|6|1.026|1.85|
> |**Qbert**|**6**|**1.670**|**15.015**|
> |**Seaquest**|**18**|**2.085**|**16.906**|
> |UpNDown|6|0.399|6.810|
>
> This reveals that the two tasks with the worst performance indeed exhibit a much greater deviation, confirming the more skewed distributions. These extreme distributions might push the exploration policy too aggressively and hinder the learning.
>
> Consider the characteristics of these tasks: they are both highly dynamic, featuring numerous moving objects and complex interactions. Such complexity makes it challenging for LLMs to obtain clear analyses. Also, the intrinsic hallucination of LLMs can cause the model to generate irrational exploration distribution in some cases.
>
> As you suggested, we design a simple safeguard mechanism to mitigate this. Before applying the generated exploration distributions, we calculate its KL divergence to uniform. If this value exceeds the running average of past KL divergences by more than 10 standard deviations, the agent discards the skewed distribution for that cycle and falls back to the default uniform exploration strategy. For each task, we average 3 repeated trainings with different random seeds:
>
> |Task|DQN|DQN+LLM-Explorer (original)||DQN+LLM-Explorer (KL clip)||
> |:-:|:-:|:-:|:-:|:-:|:-:|
> ||Score|Score|$Max[KL(p,U)]$|Score|$Max[KL(p,U)]$|
> |Qbert|306.07|301.97|1.670|312.43|1.038|
> |Seaquest|201.58|196.15|2.085|209.37|1.234|
>
> This demonstrates that such instability can be partially mitigated. Also, we believe the safeguards design you suggested is valuable for future research.
>
> **Q3. Extended baseline comparisons**
>
> We apologize for the insufficient baseline comparison. Here, we add comparisons against multiple recent adaptive exploration methods:
> - **RIDE [1].** A type of intrinsic reward that encourages the actions that changes its learned state representation.
> - **NovelD [2].** A method that weights all novel areas equally to balance the unexplored spaces.
> - **E3B [3].** A SOTA exploration method that encourages the agent to explore states that are diverse under a learned embedding.
> - **SOFE [4].** A design that expands the state space but simplifies the optimization of the agent’s objective. It can be combined with and augment frameworks like E3B.
>
> For each task, we average 3 repeated trainings with different random seeds:
>
> |Task|RND|RIDE|NovelD|E3B|E3B+SOFE|**LLM-Explorer**|
> |:-:|:-:|:-:|:-:|:-:|:-:|:-:|
> |Alien|257.53|260.26|262.74|262.38|267.19|**268.44**|
> |Freeway|10.63|10.27|16.91|17.82|18.89|**20.64**|
> |MsPacman|460.54|470.18|469.37|481.28|478.34|**489.90**|
>
> These modern techniques augment the extrinsic rewards with intrinsic rewards. In contrast, our method leverages the analyzing and reasoning capabilities of LLMs to provide context-aware guidance for the exploration. The results show that LLM-Explorer leads to the best performance.
>
> [1] RIDE: Rewarding Impact-Driven Exploration for Procedurally-Generated Environments. ICLR 2020.\
> [2] NovelD: A Simple yet Effective Exploration Criterion. NeurIPS 2021.\
> [3] Exploration via Elliptical Episodic Bonuses. Neurips 2022.\
> [4] Improving Intrinsic Exploration by Creating Stationary Objectives. ICLR 2024.
>
> **Q4. Performance of small and large LLMs**
>
> This is indeed an interesting observation. We thank the reviewer for prompting this deeper analysis, which improves the understanding of our framework.
>
> **First**, to test our hypothesis that larger models "greedily fit specific actions", we measure the entropy of the exploration distributions generated during training.
>
> |Task|Action dimension|Average Entropy||
> |:-:|:-:|:-:|:-:|
> |||GPT-4o mini|GPT-4o|
> |Alien|18|4.177|4.030|
> |Freeway|3|1.459|1.368|
> |MsPacman|9|3.063|3.000|
>
> The results show that the average entropy from GPT-4o is consistently lower. This indicates that GPT-4o indeed has a tendency to concentrate on fewer actions.
>
> **Second**, we explore ways to better leverage larger models. As suggested, we design prompt engineering to encourage more creative exploration. We add the following instruction to the end of our original prompt:
>
> ```
> Prompt: Please think creatively and consider all kinds of feasible actions. DO NOT limit the agent to a few kinds of fixed actions.
> ```
>
> With the extended prompt, we average 3 repeated trainings with different random seeds for each task:
>
> |Task|DQN|LLM-Explorer (GPT-4o mini)||LLM-Explorer (GPT-4o)||LLM-Explorer (GPT-4o with creative prompt)||
> |:-:|:-:|:-:|:-:|:-:|:-:|:-:|:-:|
> ||Score|Average entropy|Score|Average entropy|Score|Average entropy|Score|
> |Alien|245.46|268.44|4.177|249.19|4.030|255.34|4.129|
> |Freeway|5.25|20.64|1.459|19.95|1.368|20.32|1.398|
> |MsPacman|411.07|489.90|3.063|415.60|3.000|440.58|3.017|
>
> While this shows a slight positive effect, the performance improvement is limited, suggesting that simple prompting alone cannot fully overcome the model's inherent tendency.
>
> **Third**, we investigate adjusting the temperature parameter. With various temperatures, we average 3 repeated trainings with different random seeds for each task:
>
> |Task|DQN|LLM-Explorer (GPT-4o temperature=0.5)|LLM-Explorer (GPT-4o temperature=1.0)|LLM-Explorer (GPT-4o temperature=2.0)|
> |:-:|:-:|:-:|:-:|:-:|
> |Alien|245.46|243.89|249.19|235.43|
> |Freeway|5.25|15.23|19.95|4.74|
> |MsPacman|411.07|412.97|415.60|394.19|
>
> We find that there is a delicate balance. Increasing the temperature from 0.5 to 1.0, which we use now, can improve the performance. But further increasing the temperature leads to a sharp performance decline. This suggests that excessively high temperatures introduce meaningless randomness rather than beneficial, flexible exploration. This also aligns with general recommendations from providers like OpenAI.
>
> **Finally**, to observe if this counter-intuitive performance is universal. We test multiple Qwen2.5 models with varying sizes. For each task, we average 3 repeated trainings with different random seeds:
>
> |Task|DQN|LLM-Explorer (Qwen2.5-32B)|LLM-Explorer (Qwen2.5-14B)|LLM-Explorer (Qwen2.5-7B)|LLM-Explorer (Qwen2.5-3B)|
> |:-:|:-:|:-:|:-:|:-:|:-:|
> |Alien|245.46|262.84|251.41|239.18|231.98|
> |Freeway|5.25|17.61|9.11|7.79|5.01|
> |MsPacman|411.07|480.15|435.97|407.48|383.17|
>
> In the results, the performance degrades with smaller model sizes, and models with $\leq$7B parameters generally fail to work. This suggests that the "smaller is better" phenomenon is only likely to occur in certain model types and sizes comparisons (like GPT-4o vs. GPT-4o mini). Overall, larger and more capable models still tend to perform better.
>
> Thanks again for your valuable suggestions! We will include the above discussions and improvements into our final version. Please reconsider our paper and let us know if you have any further concerns.

---

> > ### Comment · Reviewer_65Q1 · 2025-08-06
> >
> > Thank you for the author’s reply. I acknowledge that introducing additional LLM-based automation through prompting can partially alleviate the limitations of manual processes. However, to what extent does the standardization of LLM prompts affect the results? Is it necessary to design specific templates? These questions warrant further analysis. In my view, the limitations of the proposed method also deserve deeper discussion, particularly regarding failure modes and safeguards.

---

> > > ### Author Response · Authors · 2025-08-06
> > > **Discussion (PART 1/4)**
> > >
> > > Thanks for your feedback! Here we provide more targeted analyses per your suggestions.
> > >
> > > **Q1. To what extent does the standardization of LLM prompts affect the results?**
> > >
> > > As we have illustrated, the LLM-Explorer with the auto-generated task descriptions performs similarly to the original one. To quantify how different the results are when using LLM prompts and manual prompts, we show [std] between different random seeds.
> > >
> > > |Environment|Task|DQN|DQN+LLM-Explorer (manual)|DQN+LLM-Explorer (auto)|
> > > |:-:|:-:|:-:|:-:|:-:|
> > > |Atari|Alien|245.46 [1.38]|268.44 [11.27]|--|
> > > ||Amidar|22.34 [1.43]|26.75 [3.85]|25.16 [3.61]|
> > > ||BankHeist|18.64 [2.22]|19.51 [2.03]|19.60 [1.94]|
> > > ||Breakout|2.67 [0.16]|2.74 [0.24]|2.71 [0.30]|
> > > ||ChopperCommand|840.63 [14.27]|868.33 [35.85]|873.98 [32.97]|
> > > ||CrazyClimber|17070.76 [638.13]|17694.35 [1240.77]|17731.42 [1310.11]|
> > > ||Freeway|5.25 [3.27]|20.64 [0.25]|19.41 [1.26]|
> > > ||Hero|1439.70 [331.19]|2689.62 [224.39]|2513.76 [201.72]|
> > > ||Jamesbond|60.48 [12.89]|77.35 [2.71]|78.13 [3.04]|
> > > ||Krull|2933.05 [38.24]|3009.12 [125.94]|3013.67 [107.28]|
> > > ||MsPacman|411.07 [10.55]|489.90 [30.55]|501.13 [41.23]|
> > > ||Pong|-15.71 [0.34]|-14.13 [0.67]|-14.07 [0.59]|
> > > ||Qbert|306.07 [14.07]|301.97 [16.00]|303.53 [15.24]|
> > > ||Seaquest|201.58 [3.56]|196.15 [16.71]|202.03 [18.68]|
> > > ||UpNDown|1370.99 [196.37]|1489.54 [170.21]|1476.72 [151.90]|
> > > |MuJoCo|HalfCheetah|1342.75 [311.16]|1363.51 [52.05]|1358.64 [45.67]|
> > > ||Hopper|293.50 [121.34]|466.80 [113.36]|447.23 [106.83]|
> > > ||Humanoid|562.18 [59.85]|568.16 [5.84]|569.57 [4.52]|
> > > ||HumanoidStandup|78272.11 [14480.19]|107257.07 [6337.44]|106735.86 [5719.21]|
> > > ||Walker2d|376.27 [48.66]|482.57 [46.82]|477.86 [45.83]|
> > > ||**Average**|**5553.39 [835.91]**|**7211.05 [427.41]**|**7175.60 [442.70]**|
> > >
> > > As the results show, the difference in the average final performance across all tasks between LLM prompts and manual prompts is not statistically significant (Welch's t-test, p > 0.1).
> > > This demonstrates that the LLM auto-generated task descriptions enable LLM-Explorer to achieve the same effect as the manual prompts, with a negligible impact on performance.

---

> > > ### Author Response · Authors · 2025-08-06
> > > **Discussion (PART 2/4)**
> > >
> > > **Q2. Is it necessary to design specific templates?**
> > >
> > > In our response above, we used a specific template for the task description (as described in the paper line 120) to ensure the auto-generated descriptions maintain a consistent format. To further investigate the role and necessity of this specific template, we conduct three additional experiments:
> > > - **Only one-shot**: We prompt the LLM to generate task descriptions by providing only the manually crafted description for the 'Alien' game as a one-shot example, without including the specific template format itself.
> > > ```
> > > Prompt: You are writing a task description for {TaskName} to support the subsequent task solving. Here is an example for the Alien game from the Atari benchmark: {TaskDescriptionExample}. Please cover all details of actions, rewards, end conditions, and goals.
> > > ```
> > >
> > > - **Only template**: We prompt the LLM to generate task descriptions by providing only the manually crafted description for the 'Alien' game as a one-shot example, without including the specific template format itself.
> > > ```
> > > Prompt: You are writing a task description for {TaskName} to support the subsequent task solving. The required format is {TaskDescriptionFormat}. Please strictly follow the format and cover all details of actions, rewards, end conditions, and goals.
> > > ```
> > >
> > > - **Zero-shot**: We prompt the LLM to generate descriptions without providing either the specific template or the one-shot example.
> > > ```
> > > Prompt: You are writing a task description for {TaskName} to support the subsequent task solving. Please cover all details of actions, rewards, end conditions, and goals.
> > > ```
> > >
> > > To ensure a timely response, we first conducted experiments on the Atari benchmark. For each task, we average the results of three repeated training runs with different random seeds.
> > >
> > > |Environment|Task|DQN|LLM-Explorer (manual)|LLM-Explorer (auto, one-shot + template)|LLM-Explorer (auto, only one-shot)|LLM-Explorer (auto, only template)|LLM-Explorer (auto, zero-shot)|
> > > |:-:|:-:|:-:|:-:|:-:|:-:|:-:|:-:|
> > > |Atari|Alien|245.46|268.44|--|--|--|--|
> > > ||Amidar|22.34|26.75|25.16|25.41|23.93|23.12|
> > > ||BankHeist|18.64|19.51|19.60|19.21|19.24|18.87|
> > > ||Breakout|2.67|2.74|2.71|2.75|2.72|2.69|
> > > ||ChopperCommand|840.63|868.33|873.98|871.25|851.45|838.71|
> > > ||CrazyClimber|17070.76|17694.35|17731.42|17709.53|17521.82|17215.43|
> > > ||Freeway|5.25|20.64|19.41|19.88|18.95|8.31|
> > > ||Hero|1439.70|2689.62|2513.76|2498.62|2351.17|1688.93|
> > > ||Jamesbond|60.48|77.35|78.13|77.58|73.81|63.29|
> > > ||Krull|2933.05|3009.12|3013.67|3021.55|2988.67|2949.34|
> > > ||MsPacman|411.07|489.90|501.13|495.74|473.55|428.18|
> > > ||Pong|-15.71|-14.13|-14.07|-14.21|-14.48|-15.35|
> > > ||Qbert|306.07|301.97|303.53|301.19|302.58|305.15|
> > > ||Seaquest|201.58|196.15|202.03|195.87|197.32|199.86|
> > > ||UpNDown|1370.99|1489.54|1476.72|1467.13|1463.15|1394.51|
> > > ||**Average**|**1761.99**|**1919.42**|**1910.51**|**1906.54**|**1876.71**|**1794.36**|
> > >
> > > The results show that using an one-shot example with a template, and using an one-shot example alone, yield similar results; both approaches allow LLM-Explorer to achieve performance comparable to using the original manual prompts. Using only the template is slightly less effective, while providing neither a template nor an example leads to a drop in performance.
> > >
> > > This analysis underscores that the structured details of actions, rewards, end conditions, and goals, as organized in our manual prompts, are crucial for LLM-Explorer's effectiveness. By checking the automated task descriptions generated under each setting, we find that providing either an one-shot example or a template is sufficient to guide the LLM to include the necessary task information. Providing a concrete, well-structured example proves to be slightly more effective than providing only an abstract template. Conversely, when the LLM is not given any reference and asked to generate a description from scratch, it cannot reliably include all the necessary information, which in turn harms the effectiveness of LLM-Explorer.

---

> > > ### Author Response · Authors · 2025-08-06
> > > **Discussion (PART 4/4)**
> > >
> > > **Failure mode 2**: As agreed by **Reviewer giQQ**, randomness in the initial state could also lead to failure of LLM-Explorer.
> > >
> > > Let's imagine an extreme task: "get to the center of the room." If in the first episode, the agent is randomly initialized on the left side, then "moving right" actions are beneficial. But if the next episode starts on the right, the LLM, being unaware of this new initial state, would still wrongly favor exploring "moving right" based on the last episode's experience.
> > >
> > > However, most tasks, including those in our benchmarks, are not as extreme as this hypothetical example. Our approach works based on two main factors:
> > > - **Structure similarity of initial states.**: While the initial positions of elements like the agent, obstacles (e.g., cars in Freeway), and enemies (e.g., ghosts in MsPacman) are randomized, their overall structure and relationships between these elements are generally consistent across episodes.
> > > - **Dynamic similarity of agent-environment interactions.** Most tasks involve long and complex processes, not just simple, few-step action sequences. The underlying dynamics of how the agent interacts with the environment are largely the same regardless of the specific starting positions.
> > >
> > > Thereby, the effective action distribution tends to have strong similarities across different episodes of the same task. LLM-Explorer can analyze these general patterns based on the task description and action-reward history to improve the exploration distribution.
> > > - **For instance**, as shown in our case study of Freeway (Figure 4), the agent may start at different positions on the bottom of road and car patterns may vary, but to achieve the goal of "reaching the top of the road", the action "moving up" is almost always beneficial, while "no-ops" is generally ineffective. The LLM can correctly infer this state-agnostic strategic bias.
> > >
> > > **Safeguard 2**: If a task truly matches the extremity of the hypothetical example we imagined above, our current LLM-Explorer framework would likely perform poorly. For such cases, the framework can be extended to incorporate a VLM, which would receive the initial state images as an additional input alongside the task descriptions.
> > > - **For example**, if the Freeway task were modified so the goal is to "reach the opposite side" and the agent could start at either the top or the bottom, the VLM would need the initial state to determine whether "moving up" or "moving down" should be prioritized.
> > >
> > > In general, for most normal tasks, the current state-agnostic design is a good choice considering computational cost, while for some extreme situations, a VLM version may be required to incorporate initial state information.We will try to find or construct some kind of tasks with extremely different initial states, such as a modified bi-directional Freeway, to validate our analyses in the final version of our paper.
> > >
> > > ## 2 Computational consumption and efficiency
> > >
> > > As discussed with **Reviewer 78Fn**, the computational consumption in interactions with LLMs would also be a drawback of our design.
> > >
> > > As we illustrated in the response, the performance degrades with smaller model sizes, and models with less than 7B parameters generally fail to work. Also, we have found in discussion with other reviewers that incorporating state information via a Vision-Language Model (VLM) would benefit the performance. However, both of these underscore the necessity of parameter size and computational consumption in LLM-Explore design.
> > >
> > > Following the suggestion of Reviewer 78Fn, we design a simple adaptive update mechanism, which reduces the interaction frequency with LLMs while maintains the performance. Also, it is possible to fine-tune a specialized LLM, for example, distilling the exploration strategy generation from a large LLM into a smaller one, to further reduce the minimum LLM size and save more computational cost. In general, valuable future works could be conducted on improving the trade-off between performance gains and resource requirements.

---

> > > ### Author Response · Authors · 2025-08-09
> > > **Authors are looking forward to the feedback**
> > >
> > > Dear reviewer,
> > >
> > > Thanks again for your insightful comments and suggestions.
> > >
> > > During the rebuttal period, we are committed to refining our work based on the suggestions and feedback provided by the reviewers. Hopefully, our discussions will solve your concerns. If so, please kindly consider raising the score as positive feedback for our efforts.
> > >
> > > It is the last hours of rebuttal, the authors are sincerely and anxiously looking forward to your feedback. Thanks again for your time; your suggestions are of great importance to us.
> > >
> > > Sincerely,
> > > The authors

---

> ### Comment · Area_Chair_iSaT · 2025-08-06
> **Discuss rebuttal**
>
> Dear Reviewer 65Q1,
>
> The author-reviewer discussion period will end soon on Aug. 8. Please read the authors' rebuttal and engage actively in discussion with the authors.
>
> AC

---

> ### Author Response · Authors · 2025-08-06
> **Discussion (PART 3/4)**
>
> **Q3. Extended discussion on limitations.**
>
> Thank you for this suggestion. Based on our discussion and incorporating feedback from other reviewers, we provide the deeper analyses to the method's limitations and potential improvements.
>
> ## 1 Failure modes and safeguards
>
> **Failure mode 1**: The LLM outputs extremely skewed distributions in some cases, which harm the training.
>
> As we discussed above, the tasks with the worst performance exhibit more skewed output distributions. These extreme distributions might push the exploration policy too aggressively and hinder the learning. This situation is likely to happen to tasks with highly dynamics and complex interactions. Also, the intrinsic hallucination of LLMs can cause the model to generate irrational exploration distribution in some cases.
>
> **Safeguard 1-1**: As we discussed above, we have considered a basic safeguard mechanism to mitigate this. If the output distributions are too skewed in the measurement of KL divergence, the agent discards the skewed distribution for that cycle and falls back to the default uniform exploration strategy.
>
> To improve this, we further test an alternative version of the above mechanism. We recognize over-skewed distributions with the same measurement of KL divergence, but when the distributions are too skewed, the agent discards the skewed distribution for that cycle and continue to use the distribution from the last cycle. For each task, we average 3 repeated trainings with different random seeds:
>
> |Task|DQN|DQN+LLM-Explorer (original)||DQN+LLM-Explorer (KL clip raw)||DQN+LLM-Explorer (KL clip new)||
> |:-:|:-:|:-:|:-:|:-:|:-:|:-:|:-:|
> ||Score|Score|$Max[KL(p,U)]$|Score|$Max[KL(p,U)]$|Score|$Max[KL(p,U)]$|
> |Qbert|306.07|301.97|1.670|312.43|1.038|316.92|1.104|
> |Seaquest|201.58|196.15|2.085|209.37|1.234|221.81|1.427|
>
> The results indicate that this new mechanism, which reuses the last valid strategy, achieves better performance. Our current method for identifying over-skewed distributions, which is based on KL divergence (as advised by **Reviewer NcaS**), is a preliminary and simple approach. Designing more sophisticated mechanisms to more reliably identify exploration distributions that are detrimental to training remains a valuable and important area for future research.
>
> **Safeguard 1-2**: Furthermore, considering that the intrinsic hallucination of LLMs can also cause the model to output an irrational exploration distribution, we design a simple mitigation strategy inspired by self-consistency work in LLM reasoning [1]. During each update cycle, we use the same prompt to have the LLM generate an exploration distribution twice. Due to the randomness introduced by the temperature setting, this process yields two different distributions. We then calculate the KL divergence to uniform for both outputs and select the distribution with the smaller KL divergence. For each task, we average the results of three repeated training runs with different random seeds.
>
> |Task|DQN|DQN+LLM-Explorer (original)||DQN+LLM-Explorer (self-consistency)||
> |:-:|:-:|:-:|:-:|:-:|:-:|
> ||Score|Score|$Max[KL(p,U)]$|Score|$Max[KL(p,U)]$|
> |Qbert|306.07|301.97|1.670|309.21|1.122|
> |Seaquest|201.58|196.15|2.085|205.12|1.360|
>
> This approach also demonstrates a slightly positive effect on performance. To better counter the negative impacts of intrinsic LLM hallucination within the LLM-Explorer framework, adapting other sophisticated methods from the field of LLM reasoning, such as self-reflection [2], presents another potential direction for future research.
>
> [1] Self-consistency improves chain of thought reasoning in language models. ICLR 2023.\
> [2] Self-Refine: Iterative refinement with self-feedback. NeurIPS 2023.

---

> ### Author Response · Authors · 2025-08-06
>
> Thanks again for your insightful comments and suggestions.
>
> Hopefully, our discussions will solve your concerns. If so, please kindly consider raising the score as positive feedback for our efforts. Also, we would be glad to further improve our paper if you have any specific suggestions for further exploration and discussion.
>
> Sincerely,\
> The authors

---

### Official Review · Reviewer_NcaS · 2025-07-05

**Clarity:** 3
**Significance:** 2
**Originality:** 3
**Rating:** 3
**Confidence:** 4

**Summary:**

This work proposes LLM-Explorer, a two-stage plug-in that periodically replaces the fixed exploration noise of a reinforcement-learning (RL) agent with a probability distribution generated by large-language models (LLMs). Every K episodes, the first LLM reads (i) a manually written task description, (ii) a uniformly subsampled action sequence (M steps) from the last episode, and (iii) the episode return, then produces a brief textual summary. A second LLM converts that summary into either a discrete action distribution (Atari) or a bias vector added to Gaussian noise (MuJoCo). This distribution is injected into the base algorithm without changing the policy or critic. On 15 Atari and 5 MuJoCo benchmarks, attaching the plug-in to several DQN variants and TD3 yields an average gain of roughly 37 % in human-normalized score. The authors claim the method is algorithm-agnostic and broadly scalable.

**Questions:**

1. Automatic task description – Is it possible to automate generation of the task description, action list, and reward schema instead of designing them manually?
2. State-aware variant – Could a vision–language model (or another lightweight state encoder) supply per-step or episode-level state information so that the exploration distribution adapts to context?
3. Compute budget – Please report wall-clock time and API cost for a 500 k-step Atari run and compare with a tuned NoisyNet baseline under the same time/$ budget.
4. Stability checks – Did training ever collapse when the LLM produced a highly skewed distribution? Reporting, for example, the maximum KL divergence to a uniform distribution during training would help.

**Ethical Concerns:**

["NO or VERY MINOR ethics concerns only"]

**Limitations:**

Yes

**Quality:**

2

**Strengths And Weaknesses:**

Strengths
- Plug-and-play design – integrates by swapping only the exploration noise; no modification of network architecture or learning code.
- Algorithm breadth – demonstrated on five RL algorithms (DQN, Double-Dueling DQN, Rainbow, CURL, TD3) across both discrete and continuous tasks.
- Ablation studies – the paper varies summary length (M) and update frequency (K) and compares with ε-greedy, NoisyNet, and RND baselines, showing each component’s effect.

Weakness
- Manual task descriptions – every environment requires a hand-written paragraph enumerating actions and reward rules; scaling to games with many actions and complex reward functions is unrealistic without automation.
- State-agnostic distribution – the LLM never sees observations; a single global bias is applied for K episodes, so the method cannot adapt to situational context and may misguide exploration in complex domains.
- Computation and cost omitted – per-episode GPT-4-level calls could be expensive, yet wall-clock time and API cost are not reported.

---

> ### Author Rebuttal · Authors · 2025-07-31
>
> We sincerely thank the reviewer for the insightful suggestions. Here we provide a point-to-point response with additional experimental results and discussions.
>
> **Q1. Automatic task description**
>
> The concern regarding the manual creation of task descriptions is an important point.
>
> Following your suggestion, we design to employ an additional LLM to automatically generate the task descriptions:
> 1. We first state the required format of the task description (line 120) and manually craft a task description for the 'Alien' game (Appendix G).
> 2. We then prompt GPT-4o mini to generate task descriptions for other environments with the one-shot example.
> ```
> Prompt: You are writing a task description for {TaskName} to support the subsequent task solving. The required format is {TaskDescriptionFormat}. Here is an example for the Alien game from the Atari benchmark: {TaskDescriptionExample}. Please strictly follow the format and cover all details of actions, rewards, end conditions, and goals.
> ```
> 3. We replace the original, hand-written task descriptions with the automatically generated ones and re-run the experiments. For each task, we average 3 repeated trainings with different random seeds:
>
> |Environment|Task|DQN|DQN+LLM-Explorer (manual)|DQN+LLM-Explorer (auto)|
> |:-:|:-:|:-:|:-:|:-:|
> |Atari|Alien|245.46|268.44|--|
> ||Amidar|22.34|26.75|25.16|
> ||BankHeist|18.64|19.51|19.60|
> ||Breakout|2.67|2.74|2.71|
> ||ChopperCommand|840.63|868.33|873.98|
> ||CrazyClimber|17070.76|17694.35|17731.42|
> ||Freeway|5.25|20.64|19.41|
> ||Hero|1439.70|2689.62|2513.76|
> ||Jamesbond|60.48|77.35|78.13|
> ||Krull|2933.05|3009.12|3013.67|
> ||MsPacman|411.07|489.90|501.13|
> ||Pong|-15.71|-14.13|-14.07|
> ||Qbert|306.07|301.97|303.53|
> ||Seaquest|201.58|196.15|202.03|
> ||UpNDown|1370.99|1489.54|1476.72|
> |MuJoCo|HalfCheetah|1342.75|1363.51|1358.64|
> ||Hopper|293.50|466.80|447.23|
> ||Humanoid|562.18|568.16|569.57|
> ||HumanoidStandup|78272.11|107257.07|106735.86|
> ||Walker2d|376.27|482.57|477.86|
> ||**Average**|**5553.39**|**7211.05**|**7175.60**|
>
> As the results indicate, the performance of LLM-Explorer using the auto-generated task descriptions is similar to the performance achieved with the original manual descriptions. This demonstrates that the process of generating task descriptions can be effectively automated without a significant drop in performance, which strengthens the scalability of our method to a broader range of tasks.
>
> **Q2. State-aware variant**
>
> We agree that incorporating state information for more context-aware exploration is highly pertinent.
>
> In line with this suggestion, we add new experiments to investigate the potential benefits of a state-aware approach. Given that the state in many RL tasks consists of image frames, we believe that a Vision-Language Model (VLM) is a more suitable choice to process complex visual information. Our experimental setup is:
> - We select Qwen2.5-VL-32B as the VLM. At the end of every $K$ episodes, we input the final state image from the last episode, along with the corresponding action sequence and episode reward, to the VLM. The final state image provides critical situational context regarding the agent's success or failure in that episode. The VLM's task is to generate the exploration probability distribution for the subsequent episodes.
> - For a controlled comparison, we also run an experiment using the text-only counterpart, Qwen2.5-32B. This model, following our original state-agnostic design, processes only the action and reward sequences to generate the exploration strategy.
>
> For each task, we average 3 repeated trainings with different random seeds:
>
> |Task|DQN|DQN+LLM-Explorer (Qwen2.5-32B)|||DQN+LLM-Explorer (Qwen2.5-VL-32B)|||
> |:-:|:-:|:-:|:-:|:-:|:-:|:-:|:-:|
> ||Score|Score|Cost ($)|Time (h)|Score|Cost ($)|Time (h)|
> |Alien|245.46|262.84|0.22|1.85|271.67|0.91|2.39|
> |Freeway|5.25|17.61|0.18|5.15|19.23|0.72|5.61|
> |MsPacman|411.07|480.15|0.25|1.87|497.21|1.04|2.50|
>
> Comparing the outcomes, the VLM-based approach shows a modest improvement in performance, confirming that additional state context can indeed refine the exploration strategy. However, this comes at the cost of a significant increase in computational overhead due to the VLM's complexity.
>
> This highlights a key trade-off. Our original state-agnostic design is intentionally lightweight. As our main experiments demonstrate, analyzing the previous action trajectory and reward feedback, in conjunction with the task definition, is sufficient to guide exploration effectively and achieve performance gains over baseline algorithms. In contrast, for users who wish to maximize performance without budget limitations, the VLM-based extension can be beneficial. Taken together, our proposed text-only LLM-Explorer remains a practical and efficient solution for general use.
>
> **Q3. Compute budget**
>
> We agree that analyzing computational time and cost is crucial for assessing the practical application potential of our method. As the reviewer suggests, we report the wall-clock time and API cost for a 500k-step run across all Atari and MuJoCo tasks. For each task, we average 3 repeated trainings with different random seeds:
>
> |Environment|Task|Input token (k)|Output token (k)|Cost ($)|Time (h)|
> |:-:|:-:|:-:|:-:|:-:|:-:|
> |Atari|Alien|1243.65|897.95|0.73|9.23|
> ||Amidar|1563.54|927.00|0.79|9.37|
> ||BankHeist|1514.98|1101.00|0.89|9.42|
> ||Breakout|3821.06|2634.00|2.15|11.02|
> ||ChopperCommand|1340.64|684.00|0.61|9.18|
> ||CrazyClimber|413.23|208.00|0.19|8.59|
> ||Freeway|366.87|231.25|0.19|8.58|
> ||Hero|1191.81|686.00|0.59|9.12|
> ||Jamesbond|2133.88|1011.00|0.93|9.64|
> ||Krull|805.92|438.00|0.38|8.85|
> ||MsPacman|1456.50|1006.10|0.82|9.36|
> ||Pong|260.83|312.00|0.23|8.57|
> ||Qbert|2263.39|1626.00|1.32|9.95|
> ||Seaquest|1389.57|768.00|0.67|9.23|
> ||UpNDown|1447.75|1011.00|0.82|9.36|
> |MuJoCo|HalfCheetah|978.16|524.98|0.46|8.96|
> ||Hopper|5693.25|3611.96|3.02|12.21|
> ||Humanoid|25387.59|7477.76|8.29|22.03|
> ||HumanoidStandup|2061.12|610.47|0.68|9.45|
> ||Walker2d|4422.74|2475.20|2.15|11.21|
> ||**Average**|**2987.82**|**1412.08**|**1.30**|**10.17**|
>
> On average, each training run consumes approximately $1.3 USD in API calls and takes about 10 hours to complete. We consider this to be an acceptable and reasonable overhead.
>
> To provide a more direct comparison with baselines under a fixed time budget, we conduct an additional experiment against NoisyNet. We count the time overhead used for the LLM API call in each update cycle of LLM-Explorer to allow the NoisyNet agent to perform additional training rollouts and updates. This ensures both methods operate within the same total wall-clock time. For each task, we average 3 repeated trainings with different random seeds:
>
> |Task|Time (h)|LLM-Explorer||NoisyNet||
> |---|---|---|---|---|---|
> |||Steps (k)|Score|Steps (k)|Score|
> |Alien|9.23|500|414.79|554|377.64|
> |Freeway|8.58|500|29.16|515|10.39|
> |MsPacman|9.36|500|1173.19|562|879.12|
>
> In comparison, even when the NoisyNet baseline is allowed to train for more iterations, LLM-Explorer maintains a clear performance advantage. This shows that our design, calling LLM once every $K$ episodes, can well balance efficiency and performance.
>
> **Q4. Stability checks**
>
> It is true that LLM-Explorer is not universally effective and, as our results show, its performance is limited in certain tasks like Qbert and Seaquest.
>
> The reviewer's suggestion to use KL divergence as a measure of distribution skewness is quite suitable for diagnosing these failures. Following this suggestion, we analyze the exploration distributions generated during training for each task. We calculate the KL divergence of each distribution relative to a uniform distribution and examine the deviation between the maximum and average KL divergence values.
>
> |Task|Action dimension|$Max[KL(p,U)]$|$\frac{Max[KL(p,U)]-Mean[KL(p,U)]}{std[KL(p,U)]}$|
> |:-:|:-:|:-:|:-:|
> |Alien|18|0.773|8.298|
> |Amidar|10|1.205|5.037|
> |BankHeist|18|2.082|12.477|
> |Breakout|4|0.436|5.985|
> |ChopperCommand|18|1.969|13.319|
> |CrazyClimber|9|1.034|10.104|
> |Freeway|3|0.58|4.458|
> |Hero|18|0.847|7.385|
> |Jamesbond|18|1.085|8.552|
> |Krull|18|1.969|11.585|
> |MsPacman|9|1.182|4.751|
> |Pong|6|1.026|1.85|
> |**Qbert**|**6**|**1.670**|**15.015**|
> |**Seaquest**|**18**|**2.085**|**16.906**|
> |UpNDown|6|0.399|6.810|
>
> This analysis reveals that the two tasks with the worst performance, Qbert and Seaquest, indeed exhibit a much greater deviation, confirming they experience more skewed distributions. These extreme distributions might push the exploration policy too aggressively in specific directions, hindering the agent's ability to learn.
>
> To mitigate this, we design and test a simple safeguard mechanism. Before applying the LLM-generated exploration distribution, we calculate its KL divergence to uniform. If this value exceeds the running average of past KL divergences by more than 10 standard deviations, the output is considered an outlier. In such cases, the agent discards the skewed distribution for that cycle and falls back to the default uniform exploration strategy of the original RL algorithm. For each task, we average 3 repeated trainings with different random seeds:
>
> |Task|DQN|DQN+LLM-Explorer (original)||DQN+LLM-Explorer (KL clip)||
> |:-:|:-:|:-:|:-:|:-:|:-:|
> ||Score|Score|$Max[KL(p,U)]$|Score|$Max[KL(p,U)]$|
> |Qbert|306.07|301.97|1.670|312.43|1.038|
> |Seaquest|201.58|196.15|2.085|209.37|1.234|
>
> This demonstrates that such instability can be partially mitigated. Also, developing more stable and LLM-guided exploration is valuable for future research.
>
> Thanks again for your valuable suggestions! We will include the above discussions and improvements into our final version. Please reconsider our paper and let us know if you have any further concerns.

---

> ### Comment · Area_Chair_iSaT · 2025-08-06
> **Discuss rebuttal**
>
> Dear Reviewer NcaS:
>
> The author-reviewer discussion period will end soon on Aug. 8. Please read the authors' rebuttal and engage actively in discussion with the authors.
>
> AC

---

### Note · Authors · 2025-08-11

Dear Reviewers and Area Chair,

Thank you for your time in the paper review and rebuttal. We are glad to receive all the insightful suggestions and experience the constructive discussions.

In the rebuttal, we have discussed around common concerns regarding:
- Automatic task description (Reviewer NcaS, 65Q1)
- State-aware variant (Reviewer NcaS, 78Fn, giQQ)
- Compute budget (Reviewer NcaS, 78Fn, giQQ)
- Stability, failure mode, and safeguards (Reviewer NcaS, 65Q1, 78Fn)
- Extended baselines (Reviewer 65Q1)
- Adaptive update interval (Reviewer 78Fn)
- Random state initialization (Reviewer giQQ)

We are pleased to see that multiple reviewers acknowledged that our efforts have addressed their concerns. Additionally, for those reviewers who may not have had time to respond yet due to time constraints, we believe we should also be able to address their major issues:
- For **Reviewer NcaS**: The reviewer's questions are mostly common ones that other reviewers also mentioned, as we listed above, and other reviewers have already agreed with our responses.
- For **Reviewer 65Q1**: We have provided extensive experimental results for the follow-up questions and added a comprehensive discussion as suggested. This discussion now integrates our further thinking as well as insights gained from discussions with other reviewers.

\
We sincerely appreciate the suggestions from the reviewers and are committed to refining our work accordingly.
Thank you again for your time and efforts in NeurIPS.

Best regards,\
The authors

---

### Decision · Program_Chairs · 2025-09-17

**Decision:**

Accept (spotlight)

**Comment:**

(a) Summary

This paper investigates the task-specific RL exploration strategy using LLM. During training, LLM analyzes the action-reward trajectory and task description for generating probability distribution to guide future policy exploration. By leveraging reasoning capability of LLM, the proposed policy exploration achieves improved average performance on Atari and Mujoco benchmarks.

(b) Strengthes
+ Using LLMs to guide exploration in RL is an interesting toptic.
+ The plug-in design of LLM-Explorer ensures broad compatibility, supporting a variety of RL algorithms across both discrete and continuous action spaces.
+ The experimental evaluation coveres multiple base RL algorithms and benchmark environments, demonstrating performance gains.
+ The proposed method is simple, modular, and the writing is easy to understand.
+ The code is available for reproducing the results.

(c) Weaknesses
- Omitting state information may restrict the method’s generality and effectiveness: the method currently uses only action-reward trajectories as input to the LLMs.
- Heavy reliance on prompt engineering: Performance depends critically on carefully crafted, environment-specific prompts. Generalizability to broader domains and the influence of the prior-knowledge of LLM are uncertain.
- Computational cost: The method relies on frequent interactions with large LLMs, which can be resource-intensive.
- Weak baselines: Only compares against NoisyNet and RND, lacking evaluation against recent state-of-the-art adaptive exploration techniques.

(d) Discussion

The authors' rebuttal addressed the reviewers' concerns on computational cost of frequent LLM access, state-aware variants, automatic task description, and extended baselines. After the rebuttal and discussion, most reviewers raised their scores.

(e) Decision

This paper proposes a novel and interesting approach for guiding exploration in RL using LLM, which is a prevalent focus in current research. The framework’s plug-in design ensures broad compatibility and practical usability, while the comprehensive ablation studies and additional analyses included in the authors’ response further validate the method’s effectiveness and flexibility. Overall, the paper offers a valuable contribution to the field and is likely to inspire further research. I therefore recommend acceptance.
~